# FINE-TUNING DIFFUSION MODELS VIA INTERMEDIATE DISTRIBUTION SHAPING

**Gautham Govind Anil**[1]    **Shaan Ul Haque**[2][*][†] **Nithish Kannen**[1][*]  **Dheeraj Nagaraj**[1][‡]
**Sanjay Shakkottai**[3]    **Karthikeyan Shanmugam**[1]
[1] Google DeepMind   [2] Georgia Institute of Technology, Atlanta   [3] University of Texas at Austin

## ABSTRACT

Diffusion models are widely used for generative tasks across domains. Given a pre-trained diffusion model, it is often desirable to fine-tune it further either to correct for errors in learning or to align with downstream applications. Towards this, we examine the effect of shaping the distribution at intermediate noise levels induced by diffusion models. First, we show that existing variants of Rejection sAmpling based Fine-Tuning (RAFT), which we unify as GRAFT, can implicitly perform KL regularized reward maximization with reshaped rewards. Motivated by this observation, we introduce P-GRAFT to shape distributions at intermediate noise levels and demonstrate empirically that this can lead to more effective fine-tuning. We mathematically explain this via a bias-variance tradeoff. Next, we look at correcting learning errors in pre-trained flow models based on the developed mathematical framework. In particular, we propose inverse noise correction, a novel algorithm to improve the quality of pre-trained flow models without explicit rewards. We empirically evaluate our methods on text-to-image(T2I) generation, layout generation, molecule generation and unconditional image generation. Notably, our framework, applied to Stable Diffusion v2, improves over policy gradient methods on popular T2I benchmarks in terms of VQAScore and shows an $8.81\%$ relative improvement over the base model. For unconditional image generation, inverse noise correction improves FID of generated images at lower FLOPs/image.

## 1    INTRODUCTION

Pre-trained generative models often require task-specific adaptations based on reward feedback - a standard strategy is to leverage RL algorithms such as Proximal Policy Optimization (PPO) (Schulman et al., 2017) with KL regularized rewards. While such methods have found great success in the context of language modeling (Bai et al., 2022; Ouyang et al., 2022), their adoption to diffusion models is not straightforward. In particular, unlike autoregressive models, marginal likelihoods required for the implementation of KL regularization in PPO are *intractable* for diffusion models. Hence, in practice, KL regularization is ignored (Black et al., 2023) or relaxations such as trajectory KL regularization (Fan et al., 2023) are considered. However, ignoring the KL term results in unstable training in large-scale settings (Deng et al., 2024), whereas using the trajectory KL constraint gives subpar results (Black et al., 2023). Further, fine-tuning with trajectory KL also results in the initial value function bias problem (Domingo-Enrich et al., 2024; Uehara et al., 2024).

Apart from policy gradient methods, recent research has also focused on fine-tuning methods based on *rejection sampling* such as RSO (Liu et al., 2023a), RAFT (Dong et al., 2023) and Reinforce-Rej (Xiong et al., 2025). Further, fine-tuning based on Best-of-N (BoN) sampling and its relation to policy gradient methods have also been explored, but in the context of autoregressive models (Amini et al., 2024; Gui et al., 2024).

Given the intractability of marginal KL for diffusion models, we explore conceptual connections between rejection sampling based fine-tuning methods and KL regularized reward maximization.

---

[*]Equal contribution
[†]Work done while at Google DeepMind
[‡]Correspondence to: dheerajnagaraj@google.com

We start by re-stating known results about equivalence between rejection sampling strategies (such as classical rejection sampling from MCMC literature and Best-of-N) and KL regularized reward maximization. We do so by unifying these strategies under a common Generalized Rejection Sampling (GRS) framework, which also subsumes additional rejection sampling strategies. More importantly, we note that fine-tuning using GRS, which we term Generalized Rejection sAmpling Fine-Tuning (GRAFT) enables *marginal KL constraint* for diffusion models, despite the *marginal likelihoods being intractable*. Building on this observation, we leverage the additional structure of diffusion (and flow) models to examine the effects of shaping the intermediate distributions induced by these models as opposed to the final data distribution. In particular, we make the following contributions:

**(a)** Leveraging properties of diffusion models, we propose Partial-GRAFT (P-GRAFT) a framework which fine-tunes *only till an intermediate denoising step* by assigning the rewards of final generations to partial, noisy generations. We show that this leads to better fine-tuning empirically and provide a mathematical justification via a bias-variance tradeoff. Empirically, we demonstrate significant quality gains across the tasks of text-to-image generation, layout generation and molecule generation.

**(b)** Motivated by P-GRAFT, we analyze the effect of learning the *initial noise distribution* of flow models. In particular, we go beyond reward-based fine-tuning and introduce Inverse Noise Correction - an adapter-based, parameter-efficient method to correct errors in pre-trained flow models even *without explicit rewards*. Intuitively, we leverage the reversibility of flow models to learn an appropriate initial noise distribution. We empirically demonstrate improved quality as well as FLOPs for unconditional image generation.

**(c)** In particular, SDv2 fine-tuned using P-GRAFT demonstrates significant improvements in VQAS-core over policy-gradient methods as well as SDXL-Base across datasets. The proposed Inverse Noise Correction strategy provides significant FID improvement at reduced FLOPs/image.

A more comprehensive list of related work can be found in Appendix A.

## 2 PRELIMINARIES

### 2.1 KL REGULARIZED RL FOR GENERATIVE MODELING

Following (Stiennon et al., 2020), we introduce the following reward maximization problem in our setting: Consider a state space $\mathcal{X}$, a reward function $r : \mathcal{X} \to \mathbb{R}$ and a reference probability measure $\bar{p}$ over $\mathcal{X}$. Let $\mathcal{P}(X)$ be the set of probability measures over $\mathcal{X}$ and $\alpha \in (0, \infty)$. Define the KL regularized reward maximization objective $R^{\text{reg}} : \mathcal{P}(\mathcal{X}) \to \mathbb{R}$ by $R^{\text{reg}}(p) = \mathbb{E}_{X \sim p}[r(X)] - \alpha \mathsf{KL}(p||\bar{p})$, where $\mathsf{KL}(\cdot||\cdot)$ is the KL divergence. RL algorithms such as PPO are then used to solve for

$$p^{\text{RL}} = \arg\sup_{p \in \mathcal{P}(\mathcal{X})} R^{\text{reg}}(p). \tag{1}$$

Using the method of Lagrangian Multipliers, we can show that $p^{\text{RL}}(x) \propto \exp(r(x)/\alpha)\bar{p}(x)$. In generative modeling literature, $\bar{p}$ is often the law of generated samples from a pre-trained model - fine-tuning is done on the model so as to sample from the tilted distribution $p^{\text{RL}}$.

### 2.2 OPTIMAL DISTRIBUTION VIA REJECTION SAMPLING:

Classical rejection sampling from the Monte Carlo literature (Thomopoulos, 2012) can be used to sample from $p^{\text{RL}}$. We note this folklore result in our setting:

**Lemma 2.1.** *Let* $r(x) \leq r_{\max}$ *for some* $r_{\max}$. *Given a sample* $Y \sim \bar{p}$, *we accept it with probability* $\mathbb{P}(\mathsf{Accept}|Y) = \exp\left(\frac{r(Y)-r_{\max}}{\alpha}\right)$. *Then, conditioned on* $\mathsf{Accept}$, $Y$ *is a sample from* $p^{\text{RL}}$.

Lemma 2.1 provides a way to obtain exact samples from $p^{\text{RL}}$. A well known challenge with this method is sample inefficiency - as often in practice, $\alpha$ is small leading to small acceptance probability. Thus, in practice, methods such as Best-of-N (BoN) which always accept a fixed fraction of samples are used. We now introduce *Generalized Rejection sAmpling Fine Tuning* (GRAFT), a framework to unify such rejection sampling approaches. More specifically, Lemma 2.3 shows that this still leads to sampling from the solution of the KL regularized reward maximization objective, but with reshaped rewards. We then discuss its utility in the context of diffusion models.

## 2.3 GRAFT: GENERALIZED REJECTION SAMPLING FINE TUNING

Assume $(X^{(i)})_{i \in [M]}$ are $M$ i.i.d. samples with law $\bar{p}$ over a space $\mathcal{X}$. Given reward function $r : \mathcal{X} \to \mathbb{R}$, let the reward corresponding to $X^{(i)}$ be $R_i := r(X^{(i)})$, the empirical distribution of $(X^{(i)})_{i \in [M]}$ be $\hat{P}_X(\cdot)$ and the empirical CDF of $(R_i)_{i \in [M]}$ be $\hat{F}_R(\cdot)$. We introduce Generalized Rejection Sampling (GRS) to accept a subset of high reward samples, $\mathcal{A} := (Y^{(j)})_{j \in [M_s]} \subseteq (X^{(i)})_{i \in [M]}$, where $Y^{(j)}$ denotes the $j^{\text{th}}$ accepted sample.

**Definition 2.2. Generalized Rejection Sampling (GRS):** Let the acceptance function $A : \mathbb{R} \times [0, 1] \times \mathcal{X} \times [0, 1] \to [0, 1]$ be such that $A$ is co-ordinate wise increasing in the first two co-ordinates. The acceptance probability of sample $i$ is $p_i := A(R_i, \hat{F}_R(R_i), X^{(i)}, \hat{P}_X)$. Draw $C_i \sim \text{Ber}(p_i) \; \forall \; i \in \{1, \ldots, M\}$, not necessarily independent of each other. Then, $X^{(i)} \in \mathcal{A}$ iff $C_i = 1$.

Definition 2.2 subsumes popular rejection sampling approaches such as RAFT and BoN. We now show that GRS implicitly performs KL regularized reward maximization with the reshaped reward $\hat{r}(\cdot)$:

**Lemma 2.3.** *The law of accepted samples under GRS (Def 2.2) given by $p(X^{(1)} = x | X^{(1)} \in \mathcal{A})$ is the solution to the following KL regularized reward maximization problem:*

$$\arg\max_{\hat{p}} \left[ \mathbb{E}_{x \sim \hat{p}} \hat{r}(x) - \alpha \text{KL}\left(\hat{p} \| \bar{p}\right) \right]; \quad \frac{\hat{r}(x)}{\alpha} := \log\left( \mathbb{E}\left[ A(r(x), \hat{F}_R(r(x)), x, \hat{P}_X) | X^{(1)} = x \right] \right)$$

*Here, the expectation is with respect to the randomness in the empirical distributions $\hat{F}_R$ and $\hat{P}_X$.*

$\hat{r}(\cdot)$ is monotonically increasing with respect to the actual reward since $A$ is an increasing function of the reward and its empirical CDF. We now instantiate GRS with commonly used variants of $A$:

$\text{Top} - \text{K}$ **Sampling:** Let the reward distribution be continuous with CDF $F(\cdot)$. We accept the top $K$ samples out of the $M$ samples based on their reward values.

$$\text{Corresponding Acceptance Function:} \quad A(r, \hat{F}_R, x, \hat{P}_X) = \begin{cases} 0 & \text{if } \hat{F}_R(r) \le 1 - \frac{K}{M} \\ 1 & \text{if } \hat{F}_R(r) > 1 - \frac{K}{M} \end{cases}$$

Lemma 2.3 shows that this acceptance function results in the reshaped reward $\hat{r}$ satisfying: $\frac{\hat{r}(x)}{\alpha} = \log\left[ \sum_{k=0}^{K-1} \binom{M-1}{k} F(r(x))^{M-k-1} (1 - F(r(x)))^k \right]$.

**Preference Rewards:** Setting $M = 2$ and $K = 1$ in the above formulation gives preference rewards, i.e., $X^{(1)}$ is accepted and $X^{(2)}$ is rejected if $r(X^{(1)}) > r(X^{(2)})$ (and vice versa). Since $F$ is an increasing function, the reward is monotonically reshaped from $r(x)$ to $\log F(r(x))$.

Varying $K$ from 1 to $M$, varies the strength of the tilt in $\text{Top} - \text{K}$ sampling. In particular, $K = M$ corresponds to $\frac{\hat{r}(x)}{\alpha} = 0$ (no tilt) and $K = 1$ corresponds to $\frac{\hat{r}(x)}{\alpha} = M \log F(r(x))$.

**Binary Rewards with De-Duplication:** Suppose $r(X) \in \{0, 1\}$ (for eg., corresponds to unstable/ stable molecules in molecule generation). De-duplication of the generated samples might be necessary to maintain diversity. Given any structure function $f$ (for eg., extracts the molecule structure from a configuration), let $N_f(X, \hat{P}_X) = |\{i : f(X^{(i)}) = f(X)\}|$, i.e, the number of copies of $X$ in the data.

$$\text{Proposed Acceptance Function:} \quad A(r, \hat{F}_R, x, \hat{P}_X) = \begin{cases} 0 & \text{if } r = 0 \\ \frac{1}{N_f(x, \hat{P}_X)} & \text{if } r = 1 \end{cases}$$

Draw $C_i \sim \text{Ber}(p_i)$ without-replacement among the duplicate/similar samples (i.e, they are marginally Bernoulli but are not independent). Thus, exactly one out of the duplicate molecules are selected almost surely. Applying Lemma 2.3, we conclude that:

$$\frac{\hat{r}(x)}{\alpha} = \begin{cases} -\infty & \text{if } r(x) = 0 \\ \log \mathbb{E}\left[ \frac{1}{N_f(x, \hat{P}_X)} | X^{(1)} = x \right] & \text{if } r(x) = 1 \end{cases}$$

We see that the shaped reward increases with diversity and with the value of the original reward. We use this in the molecule generation experiments to avoid mode collapse (Section 5.2).

## 2.4 IMPLICATIONS FOR DIFFUSION MODELS:

While specialized versions of Lemma 2.3 are known in the context of AR models (Amini et al., 2024), the result is particularly useful in the context of diffusion models. Note that given a sample $x$ along with a prompt $y$, the *marginal* likelihood $\bar{p}(x|y)$ can be easily computed for AR models. For diffusion models, we *only* have access to *conditional* likelihoods along the denoising trajectory of the diffusion process whereas $\mathsf{KL}(p||\bar{p})$ is intractable. That is, if the denoising process is run from $t_N$ to $t_0$, we have access to $\bar{p}(x_{t_i}|x_{t_{i+1}})$. A commonly used relaxation is the trajectory KL, $\mathsf{KL}(p(X_{0:T})||\bar{p}(X_{0:T}))$, which can be shown as an upper bound on the marginal KL. As discussed in (Domingo-Enrich et al., 2024), this constraint can lead to the initial value function bias problem since the KL regularization is with respect to the learned reverse process (as shown in (Domingo-Enrich et al., 2024)). It becomes necessary to learn an appropriate tilt even at time $T$. In this context, Lemma 2.3 offers a simple yet effective alternative to implicitly achieve marginal KL regularization.

Based on GRS, we propose **GRAFT: Generalized Rejection sAmpling Fine Tuning** (Algorithm 7) - given a reference model $\bar{p}$, we generate samples and perform the GRS strategy proposed in 2.2. A dataset is generated from the accepted samples and standard training is done on the generated dataset.

# 3 PARTIAL-GRAFT FOR DIFFUSION MODELS

Having established that GRAFT implicitly samples from the KL regularized reward maximization objective, we now examine methods to further improve the framework. Continuous diffusion models typically start with Gaussian noise $X_T$ at time $T$ and denoise it to the output $X_0$ via a discretized continuous time SDE. With $N$ denoising steps, the model constructs a denoising trajectory $X_{t_N} \rightarrow \ldots X_{t_i} \rightarrow \cdots \rightarrow X_{t_0}$ ($t_N = T$ and $t_0 = 0$), denoted by $X_{T:0}$. We now consider the effect of *shaping the distribution of an intermediate state $X_t$*. For the rest of the discussion, we reserve $n$ and $N$ to refer to discrete timesteps, and $t$ and $T$ for continuous time. For any $t \in [0, T]$ denote the marginal density of $X_t$ by $\bar{p}_t(x)$.

We first extend GRS to Partial Generalized Rejection Sampling (P-GRS). Let $X_t^{(1)}, \ldots, X_t^{(M)}$ be partially denoised (denoised till time $t$) samples. Let their corresponding completely denoised samples be $X_0^{(1)}, \ldots, X_0^{(M)}$. Rewards are computed using the completely denoised samples (i.e. $R_i = r(X_0^i)$ for the $i^{\text{th}}$ sample). We denote the empirical distribution of $\{X_0^{(1)}, \ldots, X_0^{(M)}\}$ by $\hat{P}_{X_0}(\cdot)$ and the empirical CDF of $\{R_1, \ldots, R_M\}$ by $\hat{F}_R(\cdot)$.

**Definition 3.1. Partial Generalized Rejection Sampling (P-GRS):** Consider an acceptance function $A : \mathbb{R} \times [0, 1] \times \mathcal{X} \times [0, 1] \rightarrow [0, 1]$ such that $A$ is co-ordinate wise increasing in the first two co-ordinates. The acceptance probability of sample $i$ is $p_i := A(R_i, \hat{F}_R(R_i), X_0^{(i)}, \hat{P}_{X_0})$. Draw $C_i \sim \text{Ber}(p_i) \ \forall i \in [M]$, not necessarily independent of each other. Then, $X_t^{(i)} \in \mathcal{A}$ iff $C_i = 1$.

**Lemma 3.2.** *The law of the accepted samples under P-GRS ( Def. 3.1) given by $p_t(X_t^{(1)} = x|X_t^{(1)} \in \mathcal{A})$ is the solution to the following Proximal Policy Optimization problem:*

$$\arg\max_{\hat{p}} \left[\mathbb{E}_{X \sim \hat{p}} \hat{r}(X) - \alpha \mathsf{KL}(\hat{p}||\bar{p}_t)\right]; \quad \frac{\hat{r}(x)}{\alpha} := \log\left(\mathbb{E}\left[A(r(X_0^{(1)}), \hat{F}_R(r(X_0^{(1)})), X_0^{(1)}, \hat{P}_X)\big|X_t^{(1)} = x\right]\right)$$

The key difference is that the reshaped reward now depends on the *expected value* of the acceptance function *given a partially denoised state $X_t$*. This tilts $\bar{p}_t$ instead of $\bar{p}_0$. It is straightforward to modify the reshaped rewards corresponding to GRS to that of P-GRS. We illustrate this by instantiating Lemma 3.2 for preference rewards, as done with GRS (Lemma 2.3) above.

**Preference rewards:** With P-GRS, $p_t(X_t^{(1)} = x|X_t^{(1)} \in \mathcal{A}) \propto \bar{p}_t(x) \exp\left(\frac{\hat{r}(x)}{\alpha}\right)$ with $\frac{\hat{r}(x)}{\alpha} = \log \mathbb{E}[F(r(X_0))|X_t = x]$.

Based on Lemma 3.2, we introduce **P-GRAFT: Partial GRAFT** (Algorithms 1 and 2). Here, fine-tuning is done on a (sampled) dataset of *partially denoised vectors* instead of fully denoised vectors. The fine-tuned model *is only trained from times $T$ to $t$, and is used for denoising from noise only till time $t$. We switch to the reference model* for further denoising. The resulting final distribution is given in Appendix D.3.1. We will now discuss the mathematical aspects of P-GRAFT and provide a justification for its improved performance.

## 3.1 A BIAS-VARIANCE TRADEOFF JUSTIFICATION FOR P-GRAFT

We analyze P-GRAFT from a bias-variance tradeoff viewpoint. Let us associate reward $r(X_0)$ with $X_t$. As argued in Lemma 3.3, variance of $r(X_0)$ conditioned on $X_t$ increases with $t$. Consequently, P-GRAFT obtains noisy rewards, seemingly making it less effective than GRAFT. However, we subsequently show that the learning problem itself *becomes easier* when $t$ is large since the score function becomes simpler (i.e, the bias reduces). Therefore, we can balance the trade-off between the two by choosing an "appropriate" intermediate time $t$ for the distributional tilt.

**Lemma 3.3.** *The expected conditional variance $\mathbb{E}[\mathsf{Var}(r(X_0)|X_t)]$ is an increasing function of $t$.*

**Example:** Consider molecule generation, where molecules are generated by a pre-trained diffusion model. The generated molecule can be stable ($r(X_0) = 1$) or unstable ($r(X_0) = 0$). Intuitively, $X_t$, for $t < T$, carries more information about $r(X_0)$ than $X_T$. We reinforce this claim empirically by giving the following illustrative statistical test. Consider the two hypotheses:

$$H_0 : r(X_0) \text{ is independent of } X_t; \qquad H_1 : r(X_0) \text{ and } X_t \text{ are dependent.}$$

---

**Algorithm 1** P-GRAFT: Training

**Input:** Trainable model $p_\theta$, Reference model $\bar{p}$, Reward function $r$, Acceptance function $A$, Number of rounds $N_S$, Intermediate timestep $N_I$

1: Initialize empty set $\mathcal{D}$
2: **for** $j = 1$ to $N_S$ **do**
3:     Generate $M$ trajectories: $X_{T:0}^{(i)} \sim \bar{p}_{T:0}; \quad i \in [M]$
4:     Obtain rewards: $r(X_0^{(i)}); \quad i \in [M]$
5:     Perform P-GRS using acceptance function $A$ on $X_{t_{N_I}}^{(i)}; \quad i \in [M]$ to get accepted samples $\mathcal{A}$
6:     Perform $\mathcal{D} \leftarrow \mathcal{D} \cup \mathcal{A}$
7: **end for**
8: Train $p_\theta$ on $\mathcal{D}$ for $t \in \{t_{N_I}, \dots, t_N\}$
9: **return** $p_\theta$

---

**Algorithm 2** P-GRAFT: Inference

**Input:** Fine tuned model $\hat{p}$, Reference model $\bar{p}$, Intermediate timestep $N_I$, Per-step denoiser DEN

1: Sample $X_T \sim \mathcal{N}(0, I)$
2: **for** $n = N - 1$ to $N_I$ **do**
3:     $X_{t_n} \leftarrow \text{DEN}(\hat{p}, X_{t_{n+1}}, t_{n+1})$
4: **end for**
5: **for** $n = N_I - 1$ to $0$ **do**
6:     $X_{t_n} \leftarrow \text{DEN}(\bar{p}, X_{t_{n+1}}, t_{n+1})$
7: **end for**
8: **return** $X_{t_0}$

---

Given $X_t$, we obtain 100 roll outs $X_0^{(i)}|X_t$ for $1 \leq i \leq 100$ and its empirical average $\hat{r}(X_t) = \sum_{i=1}^{100} r(X_0^{(i)})/100$. If $r(X_0)$ is independent of $X_t$ (under $H_0$), the law of $\hat{r}(X_t)$ is the binomial distribution $\mathsf{Bin}(100, \theta)$ with $\theta = \mathbb{P}(r(X_0) = 1)$ being the marginal probability of observing a stable molecule. We perform 1000 repetitions for the experiment above for various values of $t$ and plot the

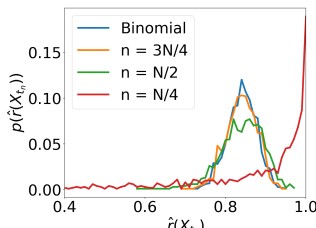

Figure 1: Law of $\hat{r}(X_t)$

Table 1: Conditional variance.

| $n$ | $\mathbb{E}\left[\mathsf{Var}(r(X_0)|X_{t_n})\right]$ |
|---|---|
| $N$ | 0.1341 |
| $3N/4$ | 0.1327 |
| $N/2$ | 0.1312 |
| $N/4$ | 0.0848 |

empirical distributions in Figure 1. For $t = t_{3N/4}$ (when $X_t$ close to $\mathcal{N}(0, \mathbf{I})$), the distribution is close to the Binomial distribution and for $t = t_{N/4}$ (when $X_t$ is close to the target) it is far. That is, $X_{t_{N/4}}$ already carries a lot of information about $r(X_0)$. This is further supported by the expected conditional variances reported in Table 1.

**Bias reduces with increasing $t$:** We follow the Stochastic Differential Equation (SDE) framework from Song et al. (2020a) for our analysis. Let the target distribution $q_0$ be the law of accepted samples under P-GRS. Diffusion models consider the forward process to be the Ornstein-Uhlenbeck Process given by $dX_t^f = -X_t^f dt + \sqrt{2}dB_t$ where $X_0^f \sim q_0$ is drawn from the target distribution over $\mathbb{R}^d$ and $B_t$ is the standard Brownian motion in $\mathbb{R}^d$. It is well known that $X_t^f \overset{d}{=} e^{-t}X_0^f + \sqrt{1 - e^{-2t}}Z$, where $Z \sim \mathcal{N}(0, \mathbf{I})$ independent of $X_0^f$.

Let $q_t$ be the density of the law of $X_t^f$. Diffusion models learn the score function $[0, T] \times \mathbb{R}^d \ni (t, X) \to \nabla \log q_t(X)$ via score matching (see Appendix A for literature review on score matching). P-GRAFT, in contrast, attempts to learn $\nabla \log q_s$ between for $s \in [t, T]$. At time $T$, $\nabla \log q_T(X) \approx -X$, the score of the standard Gaussian distribution, which is easy to learn. When $t = 0$, the score $\nabla \log q_0(X)$ corresponds to the data distribution which can be very complicated. Diffusion models use Denoising Score Matching, based on Tweedie's formula introduced by (Vincent, 2011). We show via Bakry-Emery theory (Bakry et al., 2013) that the score function $\nabla \log q_t(X)$ converges to $q_\infty(X)$ exponentially in $t$, potentially making the learning easier. Consider $s_\theta(X, t) : \mathbb{R}^d \times \mathbb{R}^+ \to \mathbb{R}^d$ to be a neural network with parameters $\theta$, then score matching objective is given by:

$$\mathcal{L}(\theta) = \mathbb{E} \int_0^T dt \| \tfrac{X_t^f - e^{-t} X_0^f}{1 - e^{-2t}} + s_\theta(X_t^f, t) \|^2.$$

In practice, $\mathcal{L}(\theta)$ is approximated with samples. By Tweedie's formula, we have: $\mathbb{E}[\tfrac{X_t^f - e^{-t} X_0^f}{1 - e^{-2t}} | X_t^f] = -\nabla \log q_t(X_t^f)$. Thus, for some constant $\mathsf{C}$, independent of $\theta$:

$$\mathcal{L}(\theta) + \mathsf{C} = \mathbb{E} \int_0^T dt \|\nabla \log q_t(X_t^f) - s_\theta(X_t^f, t)\|^2 = \int_0^T dt \int_{\mathbb{R}^d} dX \, q_t(X) \|\nabla \log q_t(X) - s_\theta(X, t)\|^2.$$

As shown by (Benton et al., 2023), $\mathcal{L}(\theta)$ directly controls the quality of generations. Note that $q_\infty$ is the density of $\mathcal{N}(0, \mathbf{I})$ and $\nabla \log q_\infty(X) = -X$. The theorem below is proved in Appendix D.5.

**Theorem 3.4.** *Define $H_t^s$ for $s \leq t$: $H_t^s = \int_s^t dt \int_{\mathbb{R}^d} dX q_s(X) \|\nabla \log q_s(X) - \nabla \log q_\infty(X)\|^2$. Then,*

$$H_t^T \leq \frac{e^{-2t}}{1 - e^{-2t}} H_0^t$$

Therefore, the score functions between time $(t, T)$ are exponentially closer to the simple Gaussian score function compared to the score functions between times $(0, t)$ in the precise sense given in Theorem 3.4. This means that the score functions at later times should be **easier** to learn.

## 4 INVERSE NOISE CORRECTION FOR FLOW MODELS

In the analysis so far, we have looked at intermediate distribution shaping in the context of KL regularized reward maximization. In particular, we have we have established bias-variance tradeoffs for diffusion models - models which use SDEs to sample from a target

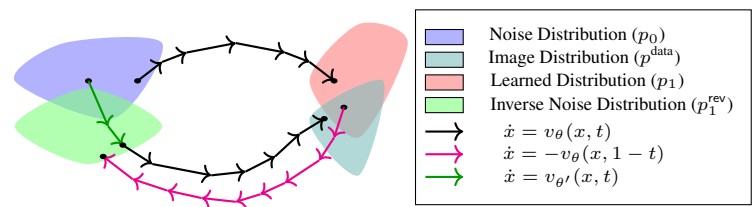

Figure 2: Inverse Noise Correction Setup

distribution. We now consider rectified flow models, which follow a deterministic ODE starting from an initial (random) noise. Hence, the final generated sample is completely determined by the initial noise. Further, the bias-variance results from the previous section indicate that the learning process is potentially easier at the initial noise level.

Since the final sample distribution is completely determined by the initial noise distribution, we ask the following question: can we correct for errors in the (learned) final distribution of the pre-trained model by correcting the initial noise distribution? Along with results from the bias-variance tradeoff discussion, we utilize another property of flow models: they admit an exact reversal. This property has been utilized extensively in the literature to map images to 'noise' for image editing Rout et al. (2024); Garibi et al. (2024) and as part of the 2-rectification Liu et al. (2022) to achieve straighter flows. We will combine these two ideas to answer the above question and develop a framework for improving flow models *even without explicit rewards*, by correcting for errors in the pre-trained distribution. We now develop this idea from first principles.

We restrict our attention to flow models with optimal transport based interpolation (Lipman et al., 2022; Liu et al., 2022), which learn a velocity field $v(x, t) : \mathbb{R}^d \times [0, 1] \to \mathbb{R}^d$ such that the following

ODE's solution at time $t = 1$ has the target distribution $p^{\text{data}}$:

$$\frac{dX_t}{dt} = v(X_t, t),\ X_0 \sim \mathcal{N}(0, \mathbf{I}). \tag{2}$$

Note that *in the literature for flow models (unlike diffusion models), $t = 0$ corresponds to noise and $t = 1$ corresponds to the target, a convention we follow in this section.*

**The errors in learned model:** Suppose we have a pre-trained vector-field, corresponding to parameter $\theta$ and solve the ODE equation 2 with $v(x, t) = v_\theta(x, t)$. Then, $\mathsf{Law}(X_1) \neq p^{\text{data}}$ due to:

**a)** Discretization error of the ODE and **b)** Statistical error due to imperfect learning.

Despite these two errors, the trained ODE is still invertible. We will leverage reversibility to arrive at our algorithm. To this end, consider the time reversal of equation 2:

$$\frac{dx_t^{\text{rev}}}{dt} = -v_\theta(x_t^{\text{rev}}, 1 - t),\ x^{\text{rev}}(0) \sim p^{\text{data}}. \tag{3}$$

---

**Algorithm 3** Inverse Noise Correction: Training

**Input:** Dataset $\mathcal{D} := \{X^{(1)}, X^{(2)}, \ldots, X^{(M)}\} \sim p^{\text{data}}$, step-size $\eta$, backward Euler steps $N_b$

1: $v_\theta = \text{TRAIN\_FLOW}(\mathcal{N}(0, \mathbf{I}), \mathcal{D})$.
2: **for** $i = 1$ to $M$ **do**
3: $\quad X_1^{\text{rev},(i)} \leftarrow \text{BWD\_Euler}(v_\theta, \eta, X^{(i)}, N_b)$
4: **end for**
5: Dataset $\mathcal{D}^{\text{rev}} \leftarrow \{X_1^{\text{rev},(1)}, \ldots, X_1^{\text{rev},(M)}\} \sim p_1^{\text{rev}}$
6: $v_{\theta^{\text{rev}}} = \text{TRAIN\_FLOW}(\mathcal{N}(0, \mathbf{I}), \mathcal{D}^{\text{rev}})$
7: **return** $v_\theta, v_{\theta'}$

---

**Algorithm 4** Inference

**Input:** Flow models $v_\theta, v_{\theta^{\text{rev}}}$, step-size $\eta$, Initial point $X_0 \sim \mathcal{N}(0, \mathbf{I})$

1: $X_1^{\text{rev}} \leftarrow \text{FWD\_Euler}(v_{\theta^{\text{rev}}}, \eta, X_0)$
2: $X_1 \leftarrow \text{FWD\_Euler}(v_\theta, \eta, X_1^{\text{rev}})$
3: **return** $X_1$

---

**The Inverse Noise:** Consider the forward Euler discretization of equation 2 with step-size $\eta$:

$$\hat{x}_{(i+1)\eta} \leftarrow \hat{x}_{i\eta} + \eta v_\theta(\hat{x}_{i\eta}, i\eta). \tag{4}$$

Let $T_{\theta,\eta}$ be the function which maps $\hat{x}_0$ to $\hat{x}_1$ i.e, $\hat{x}_1 = T_{\theta,\eta}(\hat{x}_0)$. The foward Euler approximation $T_{\theta,\eta}^{-1}(\hat{x}_1) \approx \hat{y_1}$ where $\hat{y}_{i\eta} \leftarrow \hat{y}_{(i-1)\eta} - \eta v_\theta(\hat{y}_{(i-1)\eta}, 1 - (i-1)\eta)$ with $\hat{y}_0 = \hat{x}_1$ is not good enough as noted in the image inversion/ editing literature Rout et al. (2024); Wang et al. (2024); Garibi et al. (2024). This is mitigated via numerical and control theoretic techniques. We utilize the 'backward Euler discretization' (equation 3, as used in Garibi et al. (2024)) to exactly invert equation 4.

$$\hat{x}_{\eta i}^{\text{rev}} \leftarrow \hat{x}_{\eta(i-1)}^{\text{rev}} - \eta v_\theta(\hat{x}_{\eta i}^{\text{rev}}, 1 - \eta(i-1)) \tag{5}$$

This is an implicit equation since $\hat{x}_{\eta i}^{\text{rev}}$ being calculated in the LHS also appears in the RHS. It is not apriori clear that this can be solved. Lemma 4.1 addresses this issue:

**Lemma 4.1.** *Suppose $v_\theta$ is $L$ Lipschitz in $x$ under $\ell_2$-norm and $\eta L < 1$. Then,*

*(1) $\hat{x}_{\eta i}^{\text{rev}}$ in equation 5 has a unique solution which can be obtained by a fixed point method.*

*(2) $T_{\theta,\eta}$ is invertible and $T_{\theta,\eta}^{-1}(x_0^{\text{rev}}) = x_1^{\text{rev}}$.*

That is, the mapping from noise to data given by the learned, discretized model is invertible. We show some important consequences of this in Lemma 4.2. Define the following probability distributions. Let $p^{\text{data}} = \mathsf{Law}(\text{Data})$ (i.e, target data distribution).

| $p_0 = \mathsf{Law}(\hat{x}_0) = \mathcal{N}(0, \mathbf{I})$ | $p_1 = \mathsf{Law}(\hat{x}_1)$ | $p_0^{\text{rev}} = \mathsf{Law}(\hat{x}_0^{\text{rev}}) = p^{\text{data}}$ | $p_1^{\text{rev}} = \mathsf{Law}(\hat{x}_1^{\text{rev}})$ |

We call $p_1^{\text{rev}}$ the inverse noise distribution. With perfect training and 0 discretization error, $p_1^{\text{rev}} = \mathcal{N}(0, \mathbf{I})$. However, due to these errors $p_1^{\text{rev}} \neq \mathcal{N}(0, \mathbf{I})$.

**Lemma 4.2.** *Under the assumption of Lemma 4.1, $p_1^{\text{rev}}$, $p_1$, $p^{data}$ and $p_0 = \mathcal{N}(0, \mathbf{I})$ satisfy:*

*1. $(T_{\theta,\eta})_\# p_1^{\text{rev}} = p^{data}$;    2. $\mathsf{TV}(p_1^{\text{rev}}, p_0) = \mathsf{TV}(p_1, p^{data})$;    3. $\mathsf{KL}(p_0 || p_1^{\text{rev}}) = \mathsf{KL}(p_1 || p^{data})$.*

That is, the distance between the inverse noise and the true noise is the same as the distance between the generated distribution and the true target distribution. Item 1 shows that if we can sample from the inverse noise distribution $p_1^{\mathsf{rev}}$, then we can use the pre-trained model $v_\theta(\cdot, \cdot)$ with discretization and obtain samples from the true target $p^{\mathsf{data}}$. In Kim et al. (2024), the authors note that even 2-rectification suffers when the inverse noise $p_1^{\mathsf{rev}}$ is far from $\mathcal{N}(0, \mathbf{I})$. While 2-rectification aims to improve improve the computational complexity while *maintaining quality* by aiming to obtain straight flows, we introduce inverse noise correction to *improve quality* of generations in a sample efficient way.

**Inverse Noise Correction:** Inverse Noise Correction is given in Algorithms 3 and 4, and illustrated in Figure 2. Given samples from the target distribution, $\mathcal{D}$, TRAIN_FLOW($\mathcal{N}(0, \mathbf{I}), \mathcal{D}$) trains a rectified flow model between $\mathcal{N}(0, \mathbf{I})$ to the target distribution Liu et al. (2022). Now, suppose we are given a dataset $\{X^{(1)}, \ldots, X^{(M)}\} \sim p^{\mathsf{data}}$ and a trained flow model $v_\theta$ which generates $\hat{x}_1 \sim p_1$ using equation 4 starting with $\hat{x}_0 \sim p_0$. We obtain samples $X_1^{\mathsf{rev},(i)} \sim p_1^{\mathsf{rev}}$ by backward Euler iteration in equation 5. Thereafter, we train another flow model $v_{\theta^{\mathsf{rev}}}$ which learns to sample from $p_1^{\mathsf{rev}}$ starting from $\mathcal{N}(0, \mathbf{I})$.

During inference, we sample a point from $X_0 \sim \mathcal{N}(0, \mathbf{I})$ and obtain a sample $X_1^{\mathsf{rev}} \sim p_1^{\mathsf{rev}}$ using $v_{\theta^{\mathsf{rev}}}$. Once we have the corrected noise sample, we generate images using the original flow model $v_\theta$ which now starts from $X_1^{\mathsf{rev}}$ instead of $X_0$. FWD_Euler($v_\theta, \eta, \hat{x}_0$) obtains $\hat{x}_1$ via Euler iteration (equation 4). Similarly, BWD_Euler($v_\theta, \eta, \hat{x}_0^{\mathsf{rev}}, N_b$) obtains $x_1^{\mathsf{rev}}$ by approximately solving backward Euler iteration (equation 5). They are formally described as Algorithms 5 and 6 in Appendix B. **Theoretical Justification** along the lines of Section 3.1 is given in Appendix D.8.

## 5 EXPERIMENTS

We use the notation P-GRAFT($N_I$) to denote P-GRAFT with intermediate timestep $N_I$ as described in Algorithms 1 and 2. For instance, P-GRAFT($0.75N$) would denote instantiating P-GRAFT with $N_I = 0.75N$, where $N$ is the total number of denoising steps. Recall that $t_N$ corresponds to pure noise and $t_0$ corresponds to a completely denoised sample.

### 5.1 TEXT-TO-IMAGE GENERATION

**Setup:** The objective is to fine-tune a pre-trained model so that generated images better align with prompts. We consider Stable Diffusion v2 (Rombach et al., 2022) as the pre-trained model. The reward model used is VQAScore (Lin et al., 2024) - a prompt-image alignment score between 0 to 1, with higher scores denoting better prompt-alignment. We fine-tune (separately) on GenAI-Bench (Li et al., 2024a) as well as the train split of T2ICompBench++ (Huang et al., 2025). Evaluations are done on GenAI-Bench, validation split of T2ICompBench++ and GenEval (Ghosh et al., 2023). We use LoRA (Hu et al., 2021) for compute-efficient fine-tuning. Top $-$ K sampling (Section 2.3) is used for both GRAFT and P-GRAFT. Since LoRA fine-tuning is used, the model switching in 2 can be done by simply turning off the LoRA adapter. More implementation details are given in Appendix E.

**Results:** are reported in Table 2 - for fine-tuning on GenAI-Bench, we use Top $- 10$ of $100$ samples and on T2ICompBench++, we use Top $- 1$ of $4$ samples. First, note that **both GRAFT and P-GRAFT outperform** base SDv2, SDXL-Base and DDPO. The **best performance is obtained for P-GRAFT with** $N_I = 0.25N$ across all evaluations - this clearly shows the *bias-variance tradeoff* in action. Further, both GRAFT and P-GRAFT also **generalize to unseen prompts**.

In particular, DDPO did not improve over the baseline even when trained with more samples and FLOPs as compared to GRAFT/P-GRAFT. Experiments with different sets of hyperparameters as well as adding other features such as KL regularization and a per-prompt advantage estimator on top of DDPO also did not show any significant improvements over SDv2 (see Appendix E.3). We also conduct ablations to further verify the effectiveness of the proposed methods - these include experiments on different values of $(M, K)$ in Top $-$ K of $M$ sampling, different LoRA ranks for fine-tuning as well as a reverse P-GRAFT strategy (where the fine-tuned model is used in the later denoising steps instead of initial steps). We find that P-GRAFT remains effective across different $(M, K)$ and that performance is insensitive to the LoRA rank. Further, P-GRAFT significantly outperforms reverse P-GRAFT. More details on ablations can be found in Appendix E.1.

Table 2: **Text-to-Image Generation fine-tuning on SDv2**: VQAScore (normalized to 100) reported on GenAI-Bench, T2ICompBench++ - Val (denoted as T2I - Val) and GenEval.

| Model | Fine-Tuned on GenAI-Bench | | | Fine-Tuned on T2ICompBench++ - Train | | |
|---|---|---|---|---|---|---|
| | GenAI | T2I - Val | GenEval | GenAI | T2I - Val | GenEval |
| SD v2 | $66.87_{\pm 0.14}$ | $69.20_{\pm 0.17}$ | $73.49_{\pm 0.41}$ | $66.87_{\pm 0.14}$ | $69.20_{\pm 0.17}$ | $73.49_{\pm 0.41}$ |
| SDXL-Base | $69.69_{\pm 0.17}$ | $72.98_{\pm 0.16}$ | $73.90_{\pm 0.40}$ | $69.69_{\pm 0.17}$ | $72.98_{\pm 0.16}$ | $73.90_{\pm 0.40}$ |
| DDPO | $65.70_{\pm 0.17}$ | $68.03_{\pm 0.16}$ | $72.13_{\pm 0.37}$ | $64.65_{\pm 0.17}$ | $69.05_{\pm 0.15}$ | $69.60_{\pm 0.37}$ |
| GRAFT | $70.51_{\pm 0.15}$ | $75.69_{\pm 0.13}$ | $79.85_{\pm 0.31}$ | $70.97_{\pm 0.14}$ | $75.88_{\pm 0.13}$ | $79.57_{\pm 0.30}$ |
| P-GRAFT($0.75N$) | $69.46_{\pm 0.15}$ | $74.51_{\pm 0.14}$ | $79.44_{\pm 0.33}$ | $69.51_{\pm 0.15}$ | $74.30_{\pm 0.13}$ | $78.50_{\pm 0.33}$ |
| P-GRAFT($0.5N$) | $71.00_{\pm 0.14}$ | $75.45_{\pm 0.14}$ | $80.60_{\pm 0.31}$ | $70.73_{\pm 0.14}$ | $75.37_{\pm 0.12}$ | $79.25_{\pm 0.30}$ |
| P-GRAFT($0.25N$) | $\mathbf{71.94}_{\pm 0.14}$ | $\mathbf{76.12}_{\pm 0.13}$ | $\mathbf{80.96}_{\pm 0.29}$ | $\mathbf{71.42}_{\pm 0.14}$ | $\mathbf{76.15}_{\pm 0.13}$ | $\mathbf{80.29}_{\pm 0.30}$ |

Table 3: **Layout Generation**: Fine-tuning results for unconditional and category-conditional generation on PubLayNet.

| Model | Unconditional | | Class-conditional | |
|---|---|---|---|---|
| | Alignment | FID | Alignment | FID |
| Baseline | 0.094 | 8.32 | 0.088 | 4.08 |
| GRAFT | 0.064 | 10.68 | 0.068 | 5.04 |
| P-GRAFT($0.5N$) | 0.071 | 9.24 | 0.072 | 4.55 |
| P-GRAFT($0.25N$) | **0.053** | 9.91 | **0.064** | 4.67 |

Table 4: **Molecule Generation**: Fine-tuning results on QM9. (Relative) number of sampling rounds required are also reported.

| Model | Mol: Stability | Sampling Rounds |
|---|---|---|
| Baseline | $90.50_{\pm 0.15}$ | - |
| GRAFT | $90.76_{\pm 0.20}$ | $9\times$ |
| P-GRAFT($0.5N$) | $90.46_{\pm 0.27}$ | $1\times$ |
| P-GRAFT($0.25N$) | $\mathbf{92.61}_{\pm 0.13}$ | $1\times$ |

## 5.2 LAYOUT AND MOLECULE GENERATION

**Setup:** All experiments are done on pre-trained models trained using IGD (Anil et al., 2025), a discrete-continuous diffusion framework capable of handling both layout generation and molecule generation. For layouts, we experiment with improving the alignment of elements in the generated layout as measured by the alignment metric - note that the reward is taken as 1 - alignment since lower values for the metric indicate better alignment. For molecules, the objective is to generate a larger fraction of stable molecules - molecules which are deemed stable are assigned a reward of 1 whereas unstable molecules are assigned a reward of 0. For molecule generation, *we use the de-duplication instantiation* of GRAFT/P-GRAFT (Section 2.3) to ensure diversity of generated molecules - we use RDKit to determine whether two molecules are identical or not. We use PubLayNet (Zhong et al., 2019) for layout generation, and QM9 (Ramakrishnan et al., 2014) for molecule generation. To the best of our knowledge, this is the first work which addresses fine-tuning in the context of discrete-continuous diffusion models. Ablations and experimental details are given in Appendix F.

**Results:** for layout generation are given in Table 3. Both P-GRAFT and GRAFT uniformly improve performance across both unconditional and class-conditional generation, with P-GRAFT:$0.25N$ giving the best performance. We also report FID scores computed between the generated samples and the test set of PubLayNet - this is a measure of how close the generated samples are to the pre-training distribution. As expected, the baseline has the lowest FID. Note that the FID score for P-GRAFT is smaller than GRAFT, indicating that *P-GRAFT aligns more closely* to the pre-training distribution. For molecule generation, results are given in Table 4. Again, the best performance is with P-GRAFT at $0.25N$. Note that improvement with GRAFT is marginal, despite being trained on $9\times$ the number of samples used for P-GRAFT - this points to the learning difficulty in later denoising steps.

## 5.3 IMAGE GENERATION WITH INVERSE NOISE CORRECTION

**Setup:** We consider unconditional image generation on CelebA-HQ (Karras et al., 2017) and LSUN-Church (Yu et al., 2015) at $256 \times 256$ resolution. We first train pixel-space flow models from scratch. A training corpus of inverse noise is then generated by running the trained flow models in reverse, employing the backward Euler method, on all samples in the dataset. A second flow model, which we refer to as the Noise Corrector model, is then trained to generate this inverse noise. Once the Noise Corrector is trained, this model is first used to transform standard Gaussian noise to the inverse noise. The pre-trained model then generates samples *starting from the inverse noise*. FID with 50000 generated samples with respect to the dataset is used to measure the performance. We emphasize that the our goal is not to compete with state-of-the-art (SOTA) models rather to demonstrate that

Table 5: **Image Generation**: Results for inverse noise correction on CelebA-HQ and LSUN-Church. The noise corrector samples the inverse noise starting from $\mathcal{N}(0, \mathbf{I})$ for 'Sampling Steps', and the pre-trained model samples the image starting from the inverse noise.

| Sampling Steps | | FID | | FLOPs/image $(\times 10^{12})$ |
|---|---|---|---|---|
| Noise Corrector (16M parameters) | Pre-Trained Model (65M parameters) | CelebA-HQ $(256 \times 256)$ | LSUN-Church $(256 \times 256)$ | |
| - | 1000 | 11.93 | 8.40 | 6.869 |
| - | 200 | 13.39 | 8.63 | 1.374 |
| 100 | 100 | 8.94 | 7.90 | 0.903 |
| 200 | 200 | **8.02** | **7.26** | 1.806 |

our procedure can be used to improve the performance of a given flow model by simply learning the distributional shift of noise at $t = 0$. SOTA models are larger ( Rombach et al. (2022) has $\approx 300\text{M}$ parameters) and are more sophisticated - we do not seek to match their performance.

**Results:** Table 5, shows that the Noise Corrector **significantly improves FID** scores across both datasets. Apart from quality gains, Noise Corrector also allows for **faster generation** - running the Noise Corrector for 100 steps and then running the pre-trained model for 100 steps can *outperforms* the pre-trained model with 1000 steps. The Noise Corrector only has $0.25\times$ the number of parameters, leading to further latency gains as evidenced by FLOPs counts.

## 6 CONCLUSION

We establish GRAFT, a framework for provably performing marginal KL regularized reward maximization for diffusion models through rejection sampling.We then introduce P-GRAFT, a principled framework for intermediate distribution shaping of diffusion models and provide a mathematical justification for this framework. Both GRAFT and P-GRAFT perform well empirically, outperforming policy gradient methods on the text-to-image generation task. Further, both frameworks also extend seamlessly to discrete-continuous diffusion models. Finally, we introduce Inverse Noise Correction, a strategy to improve flow models even without explicit rewards and demonstrate significant quality gains even with lower FLOPs/image.

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

# APPENDIX

## A  RELATED WORK

**Policy Gradient Methods:** Majority of the existing literature on policy gradient methods in the context of generative modeling draw inspiration from Proximal Policy Optimization(PPO) (Schulman et al., 2017) and REINFORCE (Williams and Peng, 1991). PPO based methods in the context of language modeling include Bai et al. (2022); Ouyang et al. (2022); Liu et al. (2023b); Stiennon et al. (2020), whereas frameworks based on REINFORCE include (Li et al., 2023; Ahmadian et al., 2024; Shao et al., 2024; Hu et al., 2025). Policy gradient methods have also been studied in the context of fine-tuning diffusion models (Black et al., 2023; Fan et al., 2023; Ren et al., 2024).

**Offline Fine-Tuning Methods:** Algorithms which utilize offline preference datasets for fine-tuning generative models have also been widely studied. In the context of language modeling, these include methods like SLiC (Zhao et al., 2023), DPO (Rafailov et al., 2023) and SimPO (Meng et al., 2024). Such methods have also been explore in the context of diffusion models as well - these include methods like Diffusion-DPO (Wallace et al., 2024) and Diffusion-KTO (Li et al., 2024b).

**Rejection Sampling Methods:** Recently, many works have explored rejection sampling methods in the context of autoregressive models - these include RSO (Liu et al., 2023a), RAFT (Dong et al., 2023) and Reinforce-Rej (Xiong et al., 2025). In particular, Reinforce-Rej demonstrated that rejection sampling methods can match or even outperform policy gradient methods.

**Fine-Tuning Diffusion Models:** Apart from the policy gradient methods discussed already, a host of other methods have also been proposed for fine-tuning diffusion models. Direct reward backpropagation methods include DRaFT (Clark et al., 2023) and VADER (Prabhudesai et al., 2024). Note that these methods assume access to a differentiable reward. Uehara et al. (2024) approaches the problem from the lens of entropy-regularized control - however, the method is computationally heavy and requires gradient checkpointing as well as optimizing an additional neural SDE. Domingo-Enrich et al. (2024) proposes a memoryless forward process to overcome the initial value function bias problem for the case of ODEs. PRDP Deng et al. (2024) formulates a supervised learning objective whose optimum matches with the solution to PPO, but with trajectory KL constraint - the supervised objective, with clipping, was found to make the training stable as compared to DDPO.

**Score Matching:** Score matching for distribution estimation was first introduced in (Hyvärinen and Dayan, 2005). The algorithm used in this case is called Implicit Score Matching. Diffusion models primarily use Denoising Score Matching (DSM), which is based on Tweedie's formula (Vincent, 2011; Kingma and Cun, 2010). The sample complexity of DSM has been extensively studied in the literature (Kumar et al., 2025; Block et al., 2020; Gupta et al., 2024; Chen et al., 2023).Many alternative procedures such as Sliced Score Matching (Song et al., 2020b) and Target Score Matching (De Bortoli et al., 2024) have been proposed.

**ODE Reversal in Flow Models:** A prominent use case of ODE reversal in flow models is that of image editing (Hertz et al., 2022; Kim et al., 2022; Hong et al., 2024; Mokady et al., 2023; Rout et al., 2024; Garibi et al., 2024). The reverse ODE has also been used to achieve straighter flows, allowing for faster generation, through 2-rectification/reflow algorithm (Liu et al., 2022; Lee et al., 2024; Zhu et al., 2024; Liu et al., 2023c). Notably, concurrent work Eyring et al. (2025) also proposes a strategy for aligning distilled models by fine-tuning at the noise level.

## B  ODE SOLVER ALGORITHMS

In the backward Euler Algorithm 6, at each time instant $j$ in the reverse procedure, we solve a fixed point equation to obtain high precision solution of Eq. equation 5. The step-size $\eta$ is tuned empirically so that the recursion does not blow up. Once the step-size is carefully tuned, the iteration converges to the solution at an exponential rate. In practice, we observed that $N_b = 10$ is sufficient to obtain satisfactory results.

---

**Algorithm 6** Backward Euler (BWD_Euler)

---

**Input:** Flow model $v_\theta$, step size $\eta$, sample $X^{(i)}$ from the dataset, Number of fixed point iterations $N_b$

1: $X_1^{\mathsf{rev}} = X^{(i)}$
2: **for** $j = 0$ to $\lfloor 1/\eta \rfloor - 1$ **do**
3:     $\hat{X}_0^{\mathsf{rev}} = X_j^{\mathsf{rev}}$
4:     **for** $k = 0$ to $N_b - 1$ **do**
5:        $\hat{X}_{k+1}^{\mathsf{rev}} \leftarrow X_j^{\mathsf{rev}} - \eta v_\theta(\hat{X}_k^{\mathsf{rev}}, 1 - \eta(j+1))$
6:     **end for**
7:     $X_{j+1}^{\mathsf{rev}} \leftarrow \hat{X}_{N_b}^{\mathsf{rev}}$
8: **end for**
9: **return** $X_{\lfloor 1/\eta \rfloor}^{\mathsf{rev}}$

---

**Algorithm 5** Forward Euler (FWD_Euler)

---

**Input:** Flow model $v_\theta$, step-size $\eta$, Initial point $X_0$

1: **for** $j = 0$ to $\lfloor 1/\eta \rfloor - 1$ **do**
2:     $X_{j+1} \leftarrow X_j + \eta v_\theta(X_j, \eta j)$
3: **end for**
4: **return** $X_{\lfloor 1/\eta \rfloor}$

---

## C  GRAFT: ALGORITHM

While instantiations of GRAFT are well-known in the literature and are straightforward to implement, we provide the exact algorithm here for the sake of completeness.

---

**Algorithm 7** GRAFT: Training

---

**Input:** Trainable $p_\theta$, Reference $\bar{p}$, Reward function $r$, Acceptance function $A$, Number of sampling rounds $N_S$

1: Initialize empty set $\mathcal{D}$
2: **for** $i = 0$ to $N_S$ **do**
3:     Get $M$ samples: $\{X^{(1)}, \ldots, X^{(M)}\} \sim \bar{p}$
4:     Obtain rewards: $r(X^{(i)}); \quad i \in [M]$
5:     Perform GRS using acceptance function $A$ to get accepted samples $\mathcal{A}$
6:     Perform $\mathcal{D} \leftarrow \mathcal{D} \cup \mathcal{A}$
7: **end for**
8: Train $p_\theta$ on $\mathcal{D}$
9: **return** $p_\theta$

---

# D   PROOFS

## D.1   LEMMA 2.3

*Proof.* Let $B$ be any measurable set. Consider the following probability measure:

$$\mathbb{P}(X^{(1)} \in B | X^{(1)} \in \mathcal{A}).$$

Using Bayes' rule, this measure can be rewritten as:

$$\mathbb{P}(X^{(1)} \in B | X^{(1)} \in \mathcal{A}) = \frac{\mathbb{P}(X^{(1)} \in B, X^{(1)} \in \mathcal{A})}{\mathbb{P}(X^{(1)} \in \mathcal{A})}.$$

Recall that $X^{(1)}$ is drawn from the distribution $\bar{p}$. Then, from the definition of $\mathbb{P}(X^{(1)} \in B, X^{(1)} \in \mathcal{A})$, we have:

$$\mathbb{P}(X^{(1)} \in B, X^{(1)} \in \mathcal{A}) = \int_B \mathbb{P}(X^{(1)} \in \mathcal{A} | X^{(1)} = x) d\bar{p}(x).$$

From Definition 2.2, we know that $X^{(1)} \in \mathcal{A}$ iff $C_1 = 1$. Therefore:

$$\mathbb{P}(X^{(1)} \in B, X^{(1)} \in \mathcal{A}) = \int_B \mathbb{P}(C_1 = 1 | X^{(1)} = x) d\bar{p}(x)$$

$$= \int_B \mathbb{E}\left[\mathbb{1}(C_1 = 1) | X^{(1)} = x\right] d\bar{p}(x)$$

where $\mathbb{1}(\cdot)$ denotes the indicator function. Using the tower property of expectations, this can be rewritten as:

$$\mathbb{P}(X^{(1)} \in B, X^{(1)} \in \mathcal{A}) = \int_B \mathbb{E}\left[\mathbb{E}\left[(\mathbb{1}(C_1 = 1) | X^{(1)} = x, X^{(2)}, \ldots, X^{(M)})\right]\Big| X^{(1)} = x\right] d\bar{p}(x)$$

$$= \int_B \mathbb{E}\left[\mathbb{P}\left(C_1 = 1 | X^{(1)} = x, X^{(2)}, \ldots, X^{(M)}\right)\Big| X^{(1)} = x\right] d\bar{p}(x).$$

Note that in the conditional expectation here, $X^{(1)}, \ldots, X^{(M)}$, are distributed according to $\bar{p}_0$ since $\{X^{(j)}\}_{j=1}^M$ are i.i.d samples. Again, from Definition 2.2, we know that

$$\mathbb{P}(C_1 = 1 | X^{(1)} = x, \{X^{(j)}\}_{j=2}^M) = A(r(x), \hat{F}_R(r(x)), x, \hat{P}_X)$$

where $\hat{F}_R$ and $\hat{P}_X$ are computed using the samples $\{X^{(j)}\}_{j=1}^n$. From definition of Radon-Nikodym derivative, the distribution of the accepted samples can therefore be written as:

$$\bar{p}^a(x) = Z_1 \mathbb{E}\left[A(r(x), \hat{F}_R(r(x)), x, \hat{P}_X) | X^{(1)} = x\right] \bar{p}(x) \tag{6}$$

where $Z_1 = 1/\mathbb{P}(X^{(1)} \in \mathcal{A})$ is a normalizing constant independent of $x$. Now, from the method of Lagrangian Multipliers, as mentioned in Section 2, the solution to the RL regularized reward maximization objective with reward function $\hat{r}(\cdot)$ is given by:

$$p^{\text{RL}}(x) = Z_2 \exp\left(\frac{\hat{r}(x)}{\alpha}\right) \bar{p}(x) \tag{7}$$

where $Z_2$ is the normalization constant. Comparing equation 6 and equation 7, $\bar{p}^a = p^{\text{RL}}$ whenever:

$$\frac{\hat{r}(x)}{\alpha} = \log\left(\mathbb{E}\left[A(r(x), \hat{F}_R(r(x)), x, \hat{P}_X | X^{(1)} = x\right]\right)$$

$\square$

## D.2 INSTANTIATIONS OF GRAFT

### D.2.1 TOP-$K$ OUT OF $M$ SAMPLING

Substituting $A(\cdot)$ in:

$$\log \left( \mathbb{E}_{\{X^{(j)}\}_{j=2}^n} \left[ A(r(x), \hat{F}_R(r(x)), x, \hat{P}_X) | X^{(1)} = x, \{X^{(j)}\}_{j=2}^M \right] \right)$$

we get:

$$\log \left( \int_{\{X^{(j)}\}_{j=2}^n} \mathbb{1}(r(x) \in \mathsf{Top} - \mathsf{K}(r(x), r(x^{(2)}), \ldots, r(x^{(M)}))) d\bar{p}(x^{(2)}) \ldots d\bar{p}(x^{(M)}) \right)$$

where $\mathsf{Top} - \mathsf{K}(r(x), r(x^{(2)}), \ldots, r(x^{(M)}))$ denotes the top-$K$ samples in $\{r(x), r(x^{(2)}), \ldots, r(x^{(M)})\}$. Let $U_K$ denote the event where $X^{(1)} = x$ ranks in top-$K$ among the $M$ samples, where the other $M-1$ samples are i.i.d from $\bar{p}$. This event can be decomposed as:

$$U_K = \cup_{k=1}^K E_k$$

where $E_k$ denotes the event where $r(x)$ is the $k^{\text{th}}$ in the ranked (descending) ordering of rewards. Further, note that $\{E_k\}$ are *mutually exclusive* events. Therefore:

$$\mathbb{P}(U_K) = \sum_{k=1}^K \mathbb{P}(E_k)$$

**Computing $\mathbb{P}(E_k)$:** If $x$ ranks $k^{\text{th}}$ when ranked in terms of rewards, there are $k-1$ samples which have *higher* rewards than $x$ and $M-k$ samples which have *lower* rewards than $x$. Thus, the required probability can be computed by finding the probability of having $K-1$ samples having higher rewards and the rest having lower rewards. Note that the ordering within the $K-1$ group or $M-K$ group doesn't matter. The probability of any one sample having a higher reward than $r(x)$ is $1 - F(r(x))$ and having a lower reward is $F(r(x))$. Therefore, the required probability can be computed as:

$$\mathbb{P}(E_k) = \binom{M-1}{k-1}(1 - F(r(x)))^{k-1}(F(r(x)))^{M-k}$$

And hence:

$$\mathbb{P}(U_K) = \sum_{k=0}^{K-1} \binom{M-1}{k}(1 - F(r(x)))^k(F(r(x)))^{M-k-1}$$

Therefore:

$$\frac{\hat{r}(x)}{\alpha} = \log \left( \sum_{k=0}^{K-1} \binom{M-1}{k}(1 - F(r(x)))^k(F(r(x)))^{M-k-1} \right)$$

It is straightforward to check that this is an increasing function in $r$.

### D.2.2 PREFERENCE REWARDS

Substituting $A(\cdot)$ in:

$$\log \left( \mathbb{E}_{X^{(2)}} A(r(x), \hat{F}_R(r(x)), x, \hat{P}_X) | X^{(1)} = x, X^{(2)} \right)$$

we get:

$$\log \left( \int_{X^{(2)}} \mathbb{1}(r(x^{(2)}) \leq r(x)) d\bar{p}(x^{(2)}) \right) = \log F(r(x))$$

### D.3 LEMMA 3.2

*Proof.* Let $B$ be any measurable set. Consider the following probability measure:

$$\mathbb{P}(X_t^{(1)} \in B | X_t^{(1)} \in \mathcal{A}).$$

Using Bayes' rule, this measure can be rewritten as:

$$\mathbb{P}(X_t^{(1)} \in B | X_t^{(1)} \in \mathcal{A}) = \frac{\mathbb{P}(X_t^{(1)} \in B, X_t^{(1)} \in \mathcal{A})}{\mathbb{P}(X_t^{(1)} \in \mathcal{A}))}.$$

Recall that $X_t^{(1)}$ is drawn from the distribution $\bar{p}_t$. Then, from the definition of $\mathbb{P}(X_t^{(1)} \in B, X_t^{(1)} \in \mathcal{A})$, we have:

$$\mathbb{P}(X_t^{(1)} \in B, X_t^{(1)} \in \mathcal{A}) = \int_B \mathbb{P}(X_t^{(1)} \in \mathcal{A} | X_t^{(1)} = x) d\bar{p}_t(x).$$

From Definition 3.1, we know that $X_t^{(1)} \in \mathcal{A}$ iff $C_1 = 1$. Therefore:

$$\mathbb{P}(X_t^{(1)} \in B, X_t^{(1)} \in \mathcal{A}) = \int_B \mathbb{P}(C_1 = 1 | X_t^{(1)} = x) d\bar{p}_t(x)$$
$$= \int_B \mathbb{E}\left[\mathbb{1}(C_1 = 1) | X_t^{(1)} = x\right] d\bar{p}_t(x)$$

where $\mathbb{1}(\cdot)$ denotes the indicator function. Using the tower property of expectations, this can be rewritten as:

$$\mathbb{P}(X_t^{(1)} \in B, X_t^{(1)} \in \mathcal{A}) = \int_B \mathbb{E}\left[\mathbb{E}\left[(\mathbb{1}(C_1 = 1) | X_t^{(1)} = x, X_0^{(1)}, X_0^{(2)}, \ldots, X_0^{(M)})\right] | X_t^{(1)} = x\right] d\bar{p}_t(x)$$
$$= \int_B \mathbb{E}\left[\mathbb{P}\left(C_1 = 1 | X_t^{(1)} = x, X_0^{(1)}, X_0^{(2)}, \ldots, X_0^{(M)}\right) | X_t^{(1)} = x\right] d\bar{p}_t(x).$$

Note that in the conditional expectation here, $X_0^{(2)}, \ldots, X_0^{(M)}$, are distributed according to $\bar{p}_0$ since $\{X_0^{(j)}\}_{j=1}^n$ are i.i.d samples. However, $X_0^{(1)}$ is distributed according to $\bar{p}_{0|t}$ because of the conditioning on $X_t^{(1)}$. Again, from Definition 3.1, we know that

$$\mathbb{P}(C_1 = 1 | X_t^{(1)} = x, \{X_0^{(j)}\}_{j=1}^M) = A(r(X_0^{(1)}), \hat{F}_R(r(X_0^{(1)})), X_0^{(1)}, \hat{P}_X)$$

where $\hat{F}_R$ and $\hat{P}_X$ are computed using the samples $\{X^{(j)}\}_{j=1}^M$. From the definition of Radon-Nikodym derivative, the density of the accepted samples can therefore be written as:

$$\bar{p}_t^a(x) = Z_1 \mathbb{E}\left[A(r(X_0^{(1)}), \hat{F}_R(r(X_0^{(1)})), X_0^{(1)}, \hat{P}_X | X_t^{(1)} = x)\right] \bar{p}_t(x) \tag{8}$$

where $Z_1 = 1/\mathbb{P}(X_t^{(1)} \in \mathcal{A}))$ is a normalizing constant independent of $x$. Now, from the method of Lagrangian Multipliers, as mentioned in Section 2, the solution to the RL regularized reward maximization objective (with reward function $\hat{r}(\cdot)$) is (where $Z_2$ is the normalization constant):

$$p^{\mathrm{RL}}(x) = Z_2 \exp\left(\frac{\hat{r}(x)}{\alpha}\right) \bar{p}_t(x). \tag{9}$$

Comparing equation 8 and equation 9, $\bar{p}_t^a = p^{\mathrm{RL}}$ whenever:

$$\frac{\hat{r}(x)}{\alpha} = \log\left(\mathbb{E}\left[A(r(X_0^{(1)}), \hat{F}_R(r(X_0^{(1)})), X_0^{(1)}, \hat{P}_X | X_t^{(1)} = x]\right]\right)$$

$\square$

### D.3.1 DISTRIBUTION INDUCED BY P-GRAFT

P-GRAFT uses the fine-tuned model for early denoising steps and the reference model for later denoising steps. Let us call this effective model the "stitched" model. Let us denote the distribution the stitched model samples from as $p^s$. From the discussion above, the distribution the P-GRAFT fine-tuned model samples from at time $t$ is $\bar{p}_t^a$. Further, let $\bar{p}_{0|t}$ denote the distribution of samples at time 0, given a sample at time $t$ under the reference model. Then clearly:

$$p^s(x_0) = \int \bar{p}_{0|t}(x_0|x_t)\bar{p}_t^a(x_t)dx_t$$

$$= Z \int \bar{p}_{0|t}(x_0|x_t)\bar{p}_t(x_t)\exp\left(\frac{\hat{r}(x_t)}{\alpha}\right)dx_t$$

where $Z$ is a normalization constant independent of $x_0$. In general, an explicit solution to this integral is not available.

To get more intuition regarding this distribution, we analyze two special cases. Assume that the acceptance probability depends only on $r(X_0)$ and $\hat{F}_R(r(X_0))$.

**Case 1:** The reward of a sample at time 0 is independent of the latent at time $t$, $x_t$. The acceptance probability is also independent of the latent $x_t$.

From equation 10,

$$\frac{\hat{r}(x)}{\alpha} = \log\left(\mathbb{E}\left[A(r(X_0^{(1)}), \hat{F}_R(r(X_0^{(1)})))|X_t^{(1)} = x\right]\right)$$

$$= \log\left(\mathbb{E}\left[A(r(X_0^{(1)}), \hat{F}_R(r(X_0^{(1)})))\right]\right)$$

since acceptance probability is independent of $x_t$. Note that the expectation is over $X_0^{(1)}, \ldots, X_0^{(M)}$, as discussed above. Hence, because of the independence assumption, $\log\left(\mathbb{E}\left[A(\cdot)\right]\right)$ is independent of $X_t^{(1)}$ and hence $\frac{\hat{r}(x)}{\alpha}$ is independent of $x$. Let us denote this quantity as $Z_1$. Then:

$$p^s(x_0) = Z \int \bar{p}_{0|t}(x_0|x_t)\bar{p}_t(x_t)\exp(Z_1)dx_t$$

$$= Z\exp(Z_1) \int \bar{p}_{0|t}(x_0|x_t)\bar{p}_t(x_t)dx_t$$

$$= Z\exp(Z_1)\bar{p}(x_0)$$

$$= \bar{p}(x_0)$$

where we have used the fact that $\bar{p}(x_0)$ is the normalized reference distribution. Hence, if acceptance probability of a sample at time 0 is independent of the latent $x_t$, *the stitched model results in no tilt whatsoever*.

**Case 2:** The reward of a sample at time 0 is completely determined by the latent at time t $x_t$.

A deterministic mapping $r_t$ exists such that $r(X_0) = r_t(X_t) \; \forall \; X_t$, where $X_0 \sim p_{0:t}(\cdot|X_t)$. Therefore, from equation 10,

$$\frac{\hat{r}(x)}{\alpha} = \log\left(\mathbb{E}\left[A(r(X_0^{(1)}), \hat{F}_R(r(X_0^{(1)})))|X_t^{(1)} = x\right]\right)$$

$$= \log\left(\mathbb{E}\left[A(r_t(x), \hat{F}_R(r_t(x)))\right]\right)$$

Note that the expectation here is only over $X_0^{(2)}, \ldots, X_0^{(M)}$ because of the conditioning. And hence:

$$p^s(x_0) = Z \int \bar{p}_{0|t}(x_0|x_t)\bar{p}_t(x_t)\exp\left(\frac{\hat{r}(x_t)}{\alpha}\right)dx_t$$

$$= Z \int \bar{p}_{0|t}(x_0|x_t)\bar{p}_t(x_t)\mathbb{E}\left[A(r_t(x_t), \hat{F}_R(r_t(x_t)))\right]dx_t$$

Using the fact that $r_t(x_t) = r(x_0)$, we have:

$$p^s(x_0) = Z \int \bar{p}_{0|t}(x_0|x_t)\bar{p}_t(x_t)\mathbb{E}\left[A(r(x_0), \hat{F}_R(r(x_0)))\right] dx_t$$

$$= Z\mathbb{E}\left[A(r(x_0), \hat{F}_R(r(x_0)))\right] \int \bar{p}_{0|t}(x_0|x_t)\bar{p}_t(x_t)dx_t$$

$$= Z\bar{p}(x_0)\mathbb{E}\left[A(r(x_0), \hat{F}_R(r(x_0)))\right]$$

Comparing $p^s(x_0)$ with Lemma 2.3, we see that in this case, P-GRAFT results in sampling from the exact same distribution as GRAFT.

In other cases, P-GRAFT interpolates between the reference distribution and the distribution induced by GRAFT.

## D.4 PROOF OF LEMMA 3.3

*Proof.* Let $s > t$. Note that $X_s \to X_t \to X_0$ forms a Markov chain. By the law of total variance, we have for any random variables $Y, Z$:

$$\mathsf{Var}(Z) = \mathbb{E}\mathsf{Var}(Z|Y) + \mathsf{Var}(\mathbb{E}[Z|Y])$$
$$\geq \mathbb{E}\mathsf{Var}(Z|Y) \tag{10}$$

Given $X_s$, Suppose $Z, Y$ be jointly distributed as the law of $(r(X_0), X_t)$. Then, we have $X_s$ almost surely:

$$\mathsf{Var}(r(X_0)|X_s) \geq \mathbb{E}[\mathsf{Var}(r(X_0)|X_s, X_t)|X_s] = \mathbb{E}[\mathsf{Var}(r(X_0)|X_t)|X_s] \tag{11}$$

In the last line, we have used the Markov property to show that the law of $r(X_0)|X_s, X_t$ is the same as the law of $r(X_0)|X_t$ almost surely. We conclude the result by taking expectation over both the sides. $\square$

## D.5 PROOF OF THEOREM 3.4

*Proof.* We will follow the exposition in Vempala and Wibisono (2019) for our proofs. $q_t$ converges to $q_\infty$ as $t \to \infty$. By (Vempala and Wibisono, 2019, Lemma 2) applied to the forward process, we conclude that:

$$\frac{d}{dt}\mathsf{KL}(q_t||q_\infty) = -\int_{\mathbb{R}^d} dX q_t(X)\|\nabla \log q_t(X) - \nabla \log q_\infty(X)\|^2$$

$$\implies \int_t^T dt \int_{\mathbb{R}^d} dX q_s(X)\|\nabla \log q_s(X) - \nabla \log q_\infty(X)\|^2 = \mathsf{KL}(q_t||q_\infty) - \mathsf{KL}(q_T||q_\infty) \tag{12}$$

For brevity, we call the LHS to be $H_t^T$. Clearly,

$$H_t^T - e^{-2t}H_0^T = \mathsf{KL}(q_t||q_\infty) - e^{-2t}\mathsf{KL}(q_0||q_\infty) + \mathsf{KL}(q_T||q_\infty)(e^{-2t} - 1).$$

Notice that $q_\infty$ is the density of the standard Gaussian random variable. Therefore, it satisfies the Gaussian Logarithmic Sobolev inequality Gross (1975). Thus, we can apply (Vempala and Wibisono, 2019, Theorem 4) to conclude that for every $s \geq 0$, $\mathsf{KL}(q_s||q_\infty) \leq e^{-2s}\mathsf{KL}(q_0||q_\infty)$. Thus,

$$H_t^T \leq \frac{e^{-2t}}{1 - e^{-2t}}H_0^t$$

$\square$

### D.6 PROOF OF LEMMA 4.1

The uniqueness and the convergence of fixed point iteration for implicit Euler methods have been established under great generality in Butcher (2016). However, we give a simpler proof for our specialized setting here.

1. Consider the update for the backward Euler iteration at each time step $t = \eta i$

$$\hat{x}_{\eta i}^{\mathsf{rev}} \to \hat{x}_{\eta(i-1)}^{\mathsf{rev}} - \eta v_\theta(\hat{x}_{\eta i}^{\mathsf{rev}}, 1 - \eta(i-1))$$

Let us define an operator $T_{\theta,\eta}^{\hat{x}_{\eta(i-1)}^{\mathsf{rev}}} : \mathbb{R}^d \to \mathbb{R}^d$ such that

$$T_{\theta,\eta}^{\hat{x}_{\eta(i-1)}^{\mathsf{rev}}}(x) = \hat{x}_{\eta(i-1)}^{\mathsf{rev}} - \eta v_\theta(x, 1 - \eta(i-1))$$

First, we will show that $T_{\theta,\eta}^{\hat{x}_{\eta(i-1)}^{\mathsf{rev}}}$ as defined above is a contractive operator under the condition $\eta L < 1$. Then, one can use Banach fixed point theorem to establish uniqueness of the solution and obtain the solution through fixed point iteration. To this end, consider two point $x_1$ and $x_2$ in $\mathbb{R}^d$ and apply $T_{\theta,\eta}^{\hat{x}_{\eta(i-1)}^{\mathsf{rev}}}$ to them

$$\left\| T_{\theta,\eta}^{\hat{x}_{\eta(i-1)}^{\mathsf{rev}}}(x_1) - T_{\theta,\eta}^{\hat{x}_{\eta(i-1)}^{\mathsf{rev}}}(x_2) \right\|_2 = \eta \| v_\theta(x_1, 1 - \eta(i-1)) - v_\theta(x_2, 1 - \eta(i-1)) \|_2$$
$$\leq \eta L \| x_1 - x_2 \|_2.$$

Since $\eta L < 1$, we conclude that $T_{\theta,\eta}^{\hat{x}_{\eta(i-1)}^{\mathsf{rev}}}$ is a contractive operator. Thus, by Banach fixed point theorem, the fixed point equation $T_{\theta,\eta}^{\hat{x}_{\eta(i-1)}^{\mathsf{rev}}}(x) = x$ has a unique solution for each step $t = \eta i$. To obtain the solution to the backward Euler update, we use the Banach fixed point method, i.e., start with $x_{(0)} = \hat{x}_{\eta(i-1)}^{\mathsf{rev}}$ (or any arbitrary point in $\mathbb{R}^d$) and run the iteration $x_{(k+1)} = T_{\theta,\eta}^{\hat{x}_{\eta(i-1)}^{\mathsf{rev}}}(x_{(k)})$. Then, $\lim_{k\to\infty} x_{(k)} = \hat{x}_{\eta i}^{\mathsf{rev}}$.

2. The invertibility of the operator $T_{\theta,\eta}$ follows directly from the previous part. Since the solution for the backward Euler method is unique at each time step $t = \eta i$, it implies that there exists a one-to-one mapping between sample points $x_0^{\mathsf{rev}}$ and $x_1^{\mathsf{rev}}$.

### D.7 PROOF OF LEMMA 4.2

Before starting the proof of this lemma, we will state the following well-known theorem from information theory.

**Theorem D.1.** *[Date Processing Inequality] Let $\mathcal{X}$ and $\mathcal{Y}$ be two sample spaces. Denote $\mathcal{P}(\mathcal{X})$ and $\mathcal{P}(\mathcal{Y})$ as the set of all possible probability distributions on $\mathcal{X}$ and $\mathcal{Y}$, respectively. Let $P_X, Q_X \in \mathcal{P}(\mathcal{X})$ and $P_{Y|X}$ be a transition kernel. Denote $P_Y$ and $Q_Y$ to be the push through, i.e., $P_Y(B) = \int_{\mathcal{X}} P_{Y|X}(B|X = x) dP_X(x)$. Then, for any $f$-divergence we have*

$$D_f(P_X \| Q_X) \geq D_f(P_Y | Q_Y) \tag{13}$$

1. By part 3 of Lemma 4.1, we have that $x_1^{\mathsf{rev}} = T_{\theta,\eta}^{-1}(x_0^{\mathsf{rev}}) \implies T_{\theta,\eta}(x_1^{\mathsf{rev}}) = x_0^{\mathsf{rev}}$. Suppose $x_0^{\mathsf{rev}} \sim p^*$, then by definition, $x_1^{\mathsf{rev}} \sim p_1^{\mathsf{rev}}$. This concludes the result.

2. Recall that TV-norm is an $f$-divergence. Furthermore, $T_{\theta,\eta}$ is the push forward function from $p_0$ and $p_1^{\mathsf{rev}}$ to $p_1$ and $p^*$, respectively. Thus, using DPI D.1, we have

$$\mathsf{TV}(p_1^{\mathsf{rev}}, p_0) \geq \mathsf{TV}(p^*, p_1).$$

Additionally, $T_{\theta,\eta}$ is an invertible mapping. Hence, $T_{\theta,\eta}^{-1}$ can also be viewed as the push forward function from $p_1$ and $p^*$ to $p_0$ and $p_1^{\mathsf{rev}}$, respectively. Thus, again using DPI D.1, we get

$$\mathsf{TV}(p^*, p_1) \geq \mathsf{TV}(p_1^{\mathsf{rev}}, p_0).$$

Combining both the bounds, we get the desired claim.

3. KL divergence is also a valid $f$-divergence. Thus, repeating the arguments from the previous part, one gets the desired equality.

### D.8 Theoretical Justification for Inverse Noise Correction

In this section, our goal is to provide a theoretical justification for inverse noise correction in the context of flow models. Specifically, we will argue that if $\mathsf{KL}(p^X||\mathcal{N}(0,\mathbf{I}))$ is small, then it is less challenging to learn the score function corresponding to $p_t$ and thereby the velocity field $v_t^X$ governing the rectified flow. To this end, let $X$ be a sample from a distribution $p^X$ and $Z, Y$ be standard normal random variables all independent of each other. Consider the following two linear interpolations:

$$X_t = tX + (1-t)Z \tag{14a}$$
$$Y_t = tY + (1-t)Z. \tag{14b}$$

Denote $p_t$ and $q_t$ as the distribution of $X_t$ and $Y_t$, respectively. Then, it is easy to verify that they satisfy the following continuity equations:

$$\dot{p}_t + \nabla \cdot (v_t^X p_t) = 0 \tag{15a}$$
$$\dot{q}_t + \nabla \cdot (v_t^Y q_t) = 0 \tag{15b}$$

where $v_t^X(x) = \mathbb{E}[X - Z|X_t = x]$ and $v_t^Y(x) = \mathbb{E}[Y - Z|Y_t = x]$. Then, we have the following theorem which establishes the relation between KL-divergence of $p_1$ and $q_1$ in terms of the velocities $v_t^X$ and $v_t^Y$. The proof for the theorem is provided in Section D.9.

**Theorem D.2.** *Let $p_t$ and $q_t$ be the distribution of $X_t$ and $Y_t$ defined in equation 14. Then, the KL-divergence between $p_1$ and $q_1$ satisfy the following relation*

$$\mathsf{KL}(p_1||q_1) = \mathsf{KL}(p^X||\mathcal{N}(0,\mathbf{I})) = \int_0^1 \frac{t}{1-t} \int_{\mathbb{R}^d} p_t(x) \left\| v_t^X(x) - v_t^Y(x) \right\|^2 dx dt. \tag{16}$$

Now, consider the distribution of the inverse noise $p_1^{\mathsf{rev}}$ obtained by iterating equation 5 and substitute it with $p^X$ in the theorem above. Suppose that the flow model is trained such that $\mathsf{KL}(p^{\mathsf{data}}||p_1) \leq \epsilon$. Then, by Lemma 4.2 it follows that $\mathsf{KL}(p_1^{\mathsf{rev}}||p_0) \leq \epsilon$. Combining this observation with equation 16, it is easy to see that the velocities $v_t^X(x)$ and $v_t^Y(x)$ should be close to each other. Additionally, since $q_t$ simply corresponds to learning a flow model from standard Gaussian to itself, we can explicitly compute $v_t^Y$ as follows:

$$v_t^Y(x) = \frac{x}{t} + \frac{1-t}{t} \frac{-x}{(1-t)^2 + t^2}$$
$$= \frac{x(2t-1)}{(1-t)^2 + t^2}.$$

Thus, $v_t^Y(x)$ is a linear function of $x$ and a rational function of $t$. Because $\mathsf{KL}(p_1^{\mathsf{rev}}||p_0) \leq \epsilon$, Theorem D.2 suggests that learning $v_t^X$ from data should be relatively easier as it is close to $v_t^Y$.

### D.9 Proof of Theorem D.2

Then, the time derivative of the KL-divergence between $p_t$ and $q_t$ is given by

$$\frac{d\mathsf{KL}(p_t||q_t)}{dt} = \int_{\mathbb{R}^d} \frac{d}{dt} \left( p_t(x) \log \left( \frac{p_t(x)}{q_t(x)} \right) \right) dx$$

$$= \int_{\mathbb{R}^d} \left( \dot{p}_t(x) \log \left( \frac{p_t(x)}{q_t(x)} \right) + p_t(x) \frac{d}{dt} \log(p_t(x)) - p_t(x) \frac{d}{dt} \log(q_t(x)) \right) dx$$

$$= \int_{\mathbb{R}^d} \left( \dot{p}_t(x) \log \left( \frac{p_t(x)}{q_t(x)} \right) + \dot{p}_t(x) - p_t(x) \frac{\dot{q}_t(x)}{q_t(x)} \right) dx$$

We will consider each term separately as $T_1, T_2$ and $T_3$. For $T_1$ using the continuity equation, we have

$$T_1 = \int_{\mathbb{R}^d} \dot{p}_t(x) \log \left( \frac{p_t(x)}{q_t(x)} \right) dx$$

$$= -\int_{\mathbb{R}^d} \nabla \cdot (v_t^X(x) p_t(x)) \log \left( \frac{p_t(x)}{q_t(x)} \right) dx$$

$$= \int_{\mathbb{R}^d} p_t(x) \left\langle v_t^X(x), \nabla \log \left( \frac{p_t(x)}{q_t(x)} \right) \right\rangle dx \qquad \text{(Integration by parts)}$$

Note that $\int_{\mathbb{R}^d} p_t(x)dx = 1$ for all $t \in [0, 1]$. Thus for $T_2$, we obtain

$$T_2 = \int_{\mathbb{R}^d} \dot{p}_t(x)dx = \frac{d}{dt} \int_{\mathbb{R}^d} p_t(x)dx = \frac{d}{dt} 1 = 0.$$

For the final term $T_3$, we again use the continuity equation to get

$$
\begin{aligned}
T_3 &= -\int_{\mathbb{R}^d} p_t(x)\frac{\dot{q}_t(x)}{q_t(x)}dx \\
&= \int_{\mathbb{R}^d} p_t(x)\frac{\nabla \cdot (v_t^Y q_t)}{q_t(x)}dx \\
&= -\int_{\mathbb{R}^d} q_t(x)\left\langle \nabla\left(\frac{p_t(x)}{q_t(x)}\right), v_t^Y(x)\right\rangle dx \qquad \text{(Integration by parts)} \\
&= -\int_{\mathbb{R}^d} p_t(x)\left\langle \nabla\log\left(\frac{p_t(x)}{q_t(x)}\right), v_t^Y(x)\right\rangle dx.
\end{aligned}
$$

Combining all the terms above, we get

$$\frac{d\mathsf{KL}(p_t\|q_t)}{dt} = \int_{\mathbb{R}^d} p_t(x)\left\langle \nabla\log\left(\frac{p_t(x)}{q_t(x)}\right), v_t^X(x) - v_t^Y(x)\right\rangle dx. \tag{17}$$

To obtain an expression for score function in terms of the velocity vector, we use Tweedie's formula Efron (2011) which leads us to

$$
\begin{aligned}
\mathbb{E}[X - Z|X_t = x] &= \frac{1}{1-t}\mathbb{E}[X - X_t|X_t = x] \\
&= \frac{1}{1-t}\mathbb{E}[X|X_t = x] - \frac{x}{1-t} \\
&= \frac{1}{t(1-t)}\left(x + (1-t)^2\nabla\log p_t(x)\right) - \frac{x}{1-t} \qquad \text{(Tweedie's Formula)} \\
&= \frac{x}{t} + \frac{1-t}{t}\nabla\log p_t(x). \tag{18}
\end{aligned}
$$

Similarly, we obtain

$$v_t^Y(x) = \mathbb{E}[Y - Z|Y_t = x] = \frac{x}{t} + \frac{1-t}{t}\nabla\log q_t(x). \tag{19}$$

Plugging in the expressions for the score functions into equation 17, we obtain

$$
\begin{aligned}
\frac{d\mathsf{KL}(p_t\|q_t)}{dt} &= \int_{\mathbb{R}^d} p_t(x)\left\langle \frac{t}{1-t}\left(v_t^X(x) - v_t^Y(x)\right), v_t^X(x) - v_t^Y(x)\right\rangle dx \\
&= \frac{t}{1-t}\int_{\mathbb{R}^d} p_t(x)\left\|v_t^X(x) - v_t^Y(x)\right\|^2 dx \\
\implies \mathsf{KL}(p_1\|q_1) - \mathsf{KL}(p_0\|q_0) &= \int_0^1 \frac{t}{1-t}\int_{\mathbb{R}^d} p_t(x)\left\|v_t^X(x) - v_t^Y(x)\right\|^2 dxdt.
\end{aligned}
$$

Recall that $p_0 = q_0 = q_1 = \mathcal{N}(0, \mathbf{I})$ and $p_1 = p^X$. Thus, we get the desired claim

$$\mathsf{KL}(p^X\|\mathcal{N}(0, I)) = \int_0^1 \frac{t}{1-t}\int_{\mathbb{R}^d} p_t(x)\left\|v_t^X(x) - v_t^Y(x)\right\|^2 dxdt.$$

# E  TEXT-TO-IMAGE GENERATION

## E.1  ABLATIONS

### E.1.1  DIFFERENT CHOICES OF $K$ AND $M$

We report results for various choices of $K$ and $M$ for $\mathsf{Top-K}$ of M sampling for GenAI-Bench in Tables 6, 7 and 8. Note that these models are also trained on GenAI-Bench. We also report the (mean)score separately for the "Basic" and "Advanced" split in the prompt set. Results for $\mathsf{Top-10}$ of 100 sampling for T2I-CompBench++ is given in Table 9. Models for Table 9 were trained on the train split of T2I-CompBench++. All results are consistent with the developed theory: both GRAFT and P-GRAFT outperform base SDv2 and P-GRAFT, for an appropriate choice of $N_I$ always outperform GRAFT.

Table 6: VQAScore on GenAI-Bench for $K = 1$ and $M = 4$

| Model | Basic | Advanced | Mean |
|---|---|---|---|
| SD v2 | 74.83 | 59.19 | 66.32 |
| GRAFT | 77.33 | 62.76 | 69.41 |
| P-GRAFT ($0.8N$) | 76.30 | 62.18 | 68.62 |
| P-GRAFT ($0.5N$) | 78.57 | 63.38 | **70.32** |

Table 7: VQAScore on GenAI-Bench for $K = 1$ and $M = 100$

| Model | Basic | Advanced | Mean |
|---|---|---|---|
| SD v2 | 74.83 | 59.19 | 66.32 |
| GRAFT | 79.61 | 64.26 | 71.2 |
| P-GRAFT ($0.75N$) | 76.02 | 62.91 | 68.89 |
| P-GRAFT ($0.5N$) | 78.68 | 64.5 | 70.97 |
| P-GRAFT ($0.25N$) | 80.05 | 64.85 | **71.79** |

Table 8: VQAScore on GenAI-Bench for $K = 25$ and $M = 100$

| Model | Basic | Advanced | Mean |
|---|---|---|---|
| SD v2 | 74.83 | 59.19 | 66.32 |
| GRAFT | 78.01 | 63.31 | 70.02 |
| P-GRAFT ($0.75N$) | 77.36 | 63.33 | 69.73 |
| P-GRAFT ($0.5N$) | 78.18 | 64.28 | 70.62 |
| P-GRAFT ($0.25N$) | 78.77 | 65.29 | **71.44** |

Table 9: VQAScore on T2I-CompBench++ (Val) for $K = 10$ and $M = 100$

| Model | Mean |
|---|---|
| SD v2 | 69.76 |
| GRAFT | 74.66 |
| P-GRAFT ($0.25N$) | **75.16** |

### E.1.2  CONDITIONAL VARIANCE OF REWARD FOR TEXT-TO-IMAGE GENERATION

While experimental results in Table 2 already demonstrate the bias-variance tradeoff, we provide further evidence of Lemma 3.3 in the context of text-to-image generation. We evaluate conditional variance of VQAReward scores of the base SDv2 model in GenAI-Bench. We follow the methodology as described in Section 3.1 except that we generate 4 images per prompt for a total of 1600 prompts. The results are given in Table 10. It can be seen that even at $N_I = 0.75N$, the expected conditional

variance of the reward is significantly smaller than at $t_N$. This explains why even $N_I = 0.75N$ gives a significant gain over the base model as seen in Table 2.

Table 10: Expected conditional variance for T2I generation

| $N_I$ | $\mathbb{E}\left[\text{Var}(r(X_0)|X_{t_n})\right]$ |
|---|---|
| $N$ | 0.0193 |
| $3N/4$ | 0.0080 |
| $N/2$ | 0.0039 |
| $N/4$ | 0.0019 |

### E.1.3 EFFECT OF LORA RANK

We increase the LoRA rank used for fine-tuning and check the impact on the performance. Table 11 shows that increasing LoRA rank does not seem to affect performance, indicating that the default LoRA rank is sufficient. Ablations are done on GenAI-Bench with $M = 100, K = 1$.

Table 11: Effect of LoRa Rank

| Model | Rank | Mean Reward |
|---|---|---|
| P-GRAFT ($0.5N$) | 4 | 70.97 |
| | 6 | 70.87 |
| | 8 | 70.57 |
| | 10 | 70.84 |
| P-GRAFT ($0.25N$) | 4 | 71.79 |
| | 6 | 71.84 |
| | 8 | 71.49 |
| | 10 | 71.63 |

### E.1.4 REVERSE STITCHING

In P-GRAFT, we always use the fine-tuned model for the first $(N - N_I)$ steps and then switch to the reference model. We experiment with a reverse stitching strategy, where we use the reference model for the earlier denoising steps and fine-tuned model for the later denoising steps. For switching timestep $N_I$, we denote this strategy as RP-GRAFT $(N_+I)$ - i.e. RP-GRAFT $(0.75N)$ indicates that the base model will be used from $t_N$ to $t_{0.75N}$, after which the fine-tuned model will be used. From Table 12, we observe that this strategy is significantly worse when compared to P-GRAFT - this provides further evidence of the bias-variance tradeoff. Ablations are done with $M = 100, K = 1$.

Table 12: Ablations on reverse stitching

| Model | Basic | Advanced | Mean |
|---|---|---|---|
| SDv2 | 74.83 | 59.19 | 66.32 |
| GRAFT | 79.61 | 64.26 | 71.20 |
| RP-GRAFT ($0.75N$) | 79.23 | 62.63 | 70.20 |
| RP-GRAFT ($0.5N$) | 76.60 | 60.87 | 68.05 |
| RP-GRAFT ($0.25N$) | 75.74 | 59.76 | 67.05 |

### E.2 IMPLEMENTATION DETAILS

Since we require samples only from the pre-trained model, sampling and training can be done separately. Therefore, we first perform rejection sampling according to $\text{Top} - \text{K}$ of M for the chosen values of $K$ and $M$. The selected samples are then used as the dataset for training. If not mentioned explicitly, hyperparameters can be assumed to be the default values for SD 2.0 in the Diffusers library (von Platen et al., 2022).

**Training on GenAI-Bench:**

The hyperparameters for sampling and training are given in Table 13 and Table 14 respectively. Note that one training epoch is defined as one complete pass over the training dataset. The size of the training dataset depends on the chosen $K$ and $M$. For instance, $K = 10$ and $M = 100$ results in 10 images per-prompt, for a total of 16000 images. One training epoch corresponds to a single pass over these 16000 images, which with a batch size of 8 corresponds to 2000 iterations per epoch.

Table 13: Sampling hyperparameters for GenAI-Bench

| | |
|---|---|
| Sampling Steps | 50 |
| Scheduler | EulerDiscreteScheduler |
| Guidance Scale | 7.5 |

Table 14: Training hyperparameters for GenAI-Bench

| | |
|---|---|
| Training Epochs | 10 |
| Image Resolution | $768 \times 768$ |
| Batch Size | 8 |
| Learning Rate | $10^{-4}$ |
| LR Schedule | Constant |
| LoRA Fine-Tuning | True |

**Training on T2I-CompBench++:**

The hyperparameters for sampling and training are given in Table 15 and Table 16 respectively. We use different sampling schedulers for the two datasets to ensure that our results hold irrespective of the choice of the scheduler.

Table 15: Sampling hyperparameters for T2I-CompBench++

| | |
|---|---|
| Sampling Steps | 50 |
| Scheduler | DDIMScheduler |
| $\eta$ (DDIMScheduler specific hyperparameter ) | 1.0 |
| Guidance Scale | 7.5 |

**P-GRAFT Training:**

Training and sampling using GRAFT is straightforward since standard training and inference scripts can be used out-of-the box: the only additional step need is rejection sampling on the generated samples before training. For P-GRAFT, the following changes are to be made:

- While sampling the training data, the intermediate latents should also be saved along with the final denoised image/latent. Rejection sampling is to be done on these intermediate latents, but using the rewards corresponding to the final denoised images.

- While training, note that training has to be done by noising the saved intermediate latents. This needs a re-calibration of the noise schedule, since by default, training assumes that we start from completely denoised samples. The easiest way to re-calibrate the noise schedule is by getting a new set of values for the `betas` parameter, `new_betas` as follows (where $N_I$ denotes the intermediate step of P-GRAFT):

$$\texttt{new\_betas}[0, N_I] \leftarrow 0$$
$$\texttt{new\_betas}[N_I, N] \leftarrow \texttt{betas}[N_I, N]$$

  After re-calibrating the noise, we use `new_betas` to get the corresponding `new_alphas` and `new_alphas_cumprod`. It is also necessary to note that while training, the denoiser has been trained to predict $X_0$ given any noised state $X_t$ and not the saved intermediate

Table 16: Training hyperparameters for T2I-CompBench++

| | |
|---|---|
| Training Epochs | 10 |
| Image Resolution | $768 \times 768$ |
| Batch Size | 8 |
| Learning Rate | $10^{-4}$ |
| LR Schedule | Constant |
| LoRA Fine-Tuning | True |

latent $X_{t_{N_I}}$. Let the corresponding saved completely denoised latent be $X_0$ To ensure that the training is consistent, we train using the following strategy:

Sample $\epsilon \sim \mathcal{N}(0, \mathbb{I})$

Get $X_t \leftarrow \left( \sqrt{\texttt{new\_alphas\_cumprod}[t]} \right) X_{t_{N_i}} + \left( \sqrt{1 - \texttt{new\_alphas\_cumprod}[t]} \right) \epsilon$

Get $\epsilon' \leftarrow \dfrac{X_t - \sqrt{\texttt{alphas\_cumprod}[t]} X_0}{\sqrt{1 - \texttt{alphas\_cumprod}[t]}}$

Compute Loss using $X_t$ and $\epsilon'$

### E.3 POLICY GRADIENT ALGORITHMS

DDPO(Black et al., 2023) is an on-policy policy gradient method for diffusion models that optimizes a clipped importance-weighted objective over the denoising trajectory. The original paper reports results on experiments using at most $400$ prompts. Both prompt sets we consider are significantly larger ($1600$ prompts for GenAI-Bench and $5600$ (train) prompts for T2I-CompBench++). This difference is crucial, since it has been shown in Deng et al. (2024) that scaling DDPO to large prompt sets result in unstable training and subpar performance. We also observe this phenomenon, as evidenced by the results in Table 2. As menioned in the main text, we also augment DDPO with additional elements in an attempt to improve performance. In particular, we study the following variants:

1. **DDPO:** Clipped importance-weighted policy gradient.
2. **DDPO+KL** DDPO augmented with a stepwise KL regularizer to the (frozen) reference model.
3. **DDPO+KL+EMA** DDPO with KL regularization as well as a prompt-wise exponential-moving-average baseline for advantage estimation.

**Baseline Implementation:** We use the official PyTorch implementation of DDPO[1] - we further adapt the codebase to implement other variants. Fine-tuning is always done on SDv2 using LoRA on the UNet only with a LoRA rank of 16. For the results reported in Table 2, we retain the hyperparameters used in Black et al. (2023). In particular, we use a PPO clip range of $10^{-4}$, gradient clipping norm of 1.0, Adam optimizer with $\beta_1 = 0.9, \beta_2 = 0.999$ and weight decay of $10^{-4}$. Following the original paper, we train with a relatively high learning rate of $3 \times 10^{-4}$ since LoRA fine-tuning is used. We sample 32 prompts per epoch and train with a batch size of 8, leading to 4 training iterations per epoch. However, note that each training iteration requires gradients across *the whole denoising trajectory* - this means that within each training iteration, 50 gradient calls are needed, corresponding to 50 sampling steps. For GenAI-Bench, training is done for 500 such epochs, whereas for T2I-Compbench++, training is done for 800 epochs. With this setup, in Tables 17 and 18, we compare the sampling/compute requirements for DDPO and GRAFT/P-GRAFT. In particular, note that GRAFT/P-GRAFT already outperforms DDPO with $K = 1, M = 4$ despite DDPO being trained on $10\times$ more samples and $50\times$ more gradient calls.

**Additional configurations with base hyperparameters:** With the base hyperparameters described above, we also try augmenting DDPO with KL and EMA as described above. The training curves are given in Figure 3.

---

[1] https://github.com/kvablack/ddpo-pytorch

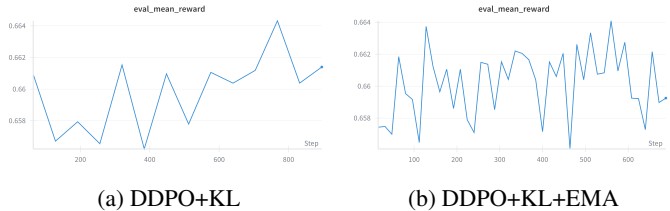

(a) DDPO+KL            (b) DDPO+KL+EMA

Figure 3: Training curves for the three policy-gradient baselines on GenAI Bench (1,600 prompts) with low value of clipping.

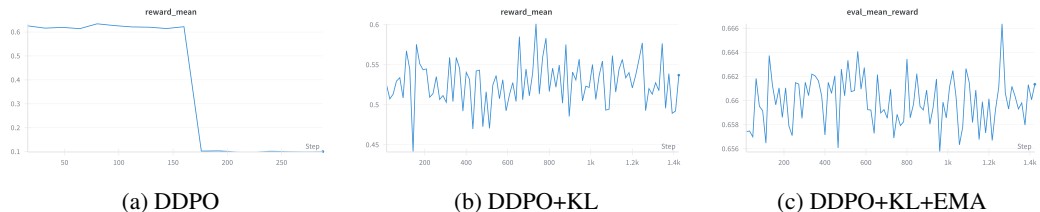

(a) DDPO        (b) DDPO+KL        (c) DDPO+KL+EMA

Figure 4: Training curves for the three policy-gradient baselines on GenAI Bench (1,600 prompts) with high value of clipping.

**DDPO:** We also try additional settings for hyperparameters apart from the oens we have reported so far. Sampling uses DDIM with $T \in [40, 50]$ steps and classifier-free guidance $g = 5$. Optimization uses AdamW with learning rates $\{2 \times 10^{-5}, 10^{-5}\}$, batch sizes $8/8$ (sampling/training), PPO-style clipping $\epsilon \in \{0.1, 0.2\}$. Following DDPO, we replay the scheduler to compute per-step log-probabilities on the same trajectories: $\ell_t = \log p_\theta(x_{t-1} \mid x_t, c)$ and $\ell_t^{\text{old}} = \log p_{\theta_0}(x_{t-1} \mid x_t, c)$. We use the clipped objective:

$$\mathcal{L}_{\text{DDPO}} = -\mathbb{E}\big[\min\big(r_t A, \ \text{clip}(r_t, 1-\epsilon, 1+\epsilon)\, A\big)\big], \qquad r_t = \exp\big(\ell_t - \ell_t^{\text{old}}\big), \qquad (20)$$

with a centered batchwise advantage $A$. Specifically, we experiment with higher clipping range, $\epsilon \in \{0.1, 0.2\}$ and use a whitened batchwise advantage. $\ell_t, \ell_t^{\text{old}}$ are obtained by replaying the DDIM scheduler on the same trajectory.

Result. On 1600 prompts, the learning curve exhibits a short initial rise followed by a sharp collapse after $\sim 150$ steps (Fig. 4). The setting of 1600 heterogeneous prompts induces high variance and many ratios $r_t$ saturate at the clipping boundary, producing low-magnitude effective gradients and the observed drop in reward.

**DDPO+KL:** We augment equation 20 with a per-step quadratic penalty to the frozen reference:

$$\mathcal{L}_{\text{DDPO+KL}} = \mathcal{L}_{\text{DDPO}} + \beta \frac{1}{T} \sum_{t=1}^{T} \big(\ell_t - \ell_t^{\text{old}}\big)^2, \qquad \beta \in \{0.02, \ 0.005\}. \qquad (21)$$

Result. The KL term prevents divergence of the policy and eliminates the reward collapse after the first few steps. Even with this, average reward improvements remain limited. Larger $\beta$ contracts the policy towards the reference, whereas smaller $\beta$ provides insufficient variance control, yielding small net gains.

**DDPO+KL+EMA (prompt-wise baseline):** To mitigate cross-prompt bias, we maintain for each prompt $z$, an EMA of reward and variance,

$$b(z) \leftarrow (1-\alpha)b(z) + \alpha\, r, \qquad v(z) \leftarrow (1-\alpha)v(z) + \alpha\big(r - b(z)\big)^2,$$

and employ a whitened advantage inside equation 20: $\widehat{A} = \dfrac{r - b(z)}{\sqrt{v(z) + \varepsilon}} + \eta, \ \eta \sim \mathcal{N}(0, \sigma^2)$.

Result. Training is the most stable among the three variants and exhibits smooth reward trajectories without collapse, yet the absolute improvement in mean reward is modest relative to the base policy.

**PRDP:** We also tried implementing PRDP (Deng et al., 2024) using the PRDP loss function provided in the appendix of the paper since no official code was provided. However, we did not see any significant improvement compared to the baseline despite following the algorithm and hyperparameters closely. One potential reason for this could be that we use LoRA fine-tuning whereas the original paper uses full fine-tuning. Further, we rely on gradient checkpointing for the implementation as well since the backpropagation is through the entire sampling trajectory.

Table 17: Comparison of Sampling Cost and Training Cost for GenAI-Bench

| Algorithm | Samples generated | Samples Trained on | Gradient Calls |
|---|---|---|---|
| GRAFT($K = 10, M = 100$) | 160k | 16k | 20k |
| GRAFT ($K = 1, M = 4$) | 6.4k | 1.6k | 2k |
| DDPO | 16k | 16k | 100k |

Table 18: Comparison of Sampling Cost and Training Cost for T2I-CompBench++

| Algorithm | Samples generated | Samples Trained on | Gradient Calls |
|---|---|---|---|
| GRAFT ($K = 1, M = 4$) | 22.4k | 5.6k | 7k |
| DDPO | 25.6k | 25.6k | 160k |

### E.4 COMPUTE FLOPs ANALYSIS OF P-GRAFT

We compare the compute cost of P-GRAFT and DDPO in terms of total UNet FLOPs. Let $F_u$ denote the cost of one UNet forward pass at $64{\times}64$ latent resolution. Following Kaplan et al. (2020), we approximate a backward training step 2 times of a forward step. So if $F_u$ is the forward step compute, a forward + backward step will incur $3F_u$. We assume a batch size of 1 for both algorithms for standardization.

For $P$ prompts, $M$ samples per prompt, top-$K$ retained, $T$ diffusion steps, $E_{\text{sft}}$ epochs. For the implementation we use the standard stable diffusion training script that only samples a single timestep $t \in [0, T]$ during training:

$$F_{\text{P-GRAFT}} = \underbrace{P\,M\,T}_{\text{sampling}} F_u \; + \; \underbrace{E_{\text{sft}}\,P\,K}_{\text{training}} 3F_u,$$

$$F_{\text{DDPO}} = \underbrace{E_{\text{ddpo}}\,N_{\text{gen}}\,T}_{\text{trajectories}} \cdot (1 + 3)F_u$$

**GenAI-Bench configuration.** We use $P{=}1600$, $M{=}100$, $K{=}10$, $T{=}40$, $E_{\text{sft}}{=}10$ for P-GRAFT; Trajectories generated per epoch $N_{\text{gen}}{=}128$, and Number of Epochs $E_{\text{ddpo}}{=}50$ for DDPO

Table 19: FLOPs in units of forward pass $F_u$ for GenAI-Bench.

| Algorithm | Sampling | Training | Total |
|---|---|---|---|
| P-GRAFT ($K{=}10$, $M{=}100$) | 6.40M | 0.48M | 6.88M |
| P-GRAFT ($K{=}1$, $M{=}4$) | 0.256M | 0.048M | 0.304M |
| DDPO ($E{=}50$, $N_{\text{gen}}{=}128$) | 0.256M | 0.768M | 1.024M |

**Discussion.** P-GRAFT's total compute is dominated by sample generation, while backpropagation is confined to fine-tuning on the selected top-$K$ samples. In contrast, DDPO backpropagates through all $T$ denoising steps online for every sample, creating a sequential bottleneck. Consequently, despite DDPO's nominal FLOPs appearing comparable or lower in our regime, its wall-clock time is substantially longer due to stepwise backward passes that are less parallelizable. Moreover, as shown in Table 2, P-GRAFT achieves higher rewards under the reported budgets; and in the compute-matched case ($K{=}1$; Table 6), P-GRAFT still outperforms DDPO, indicating that gains come from improved optimization and not just additional training compute.

## E.5 QUALITATIVE EXAMPLES

### E.5.1 GENAI-BENCH

**Prompt:** *Three flowers in the meadow, with only the red rose blooming; the others are not open.*

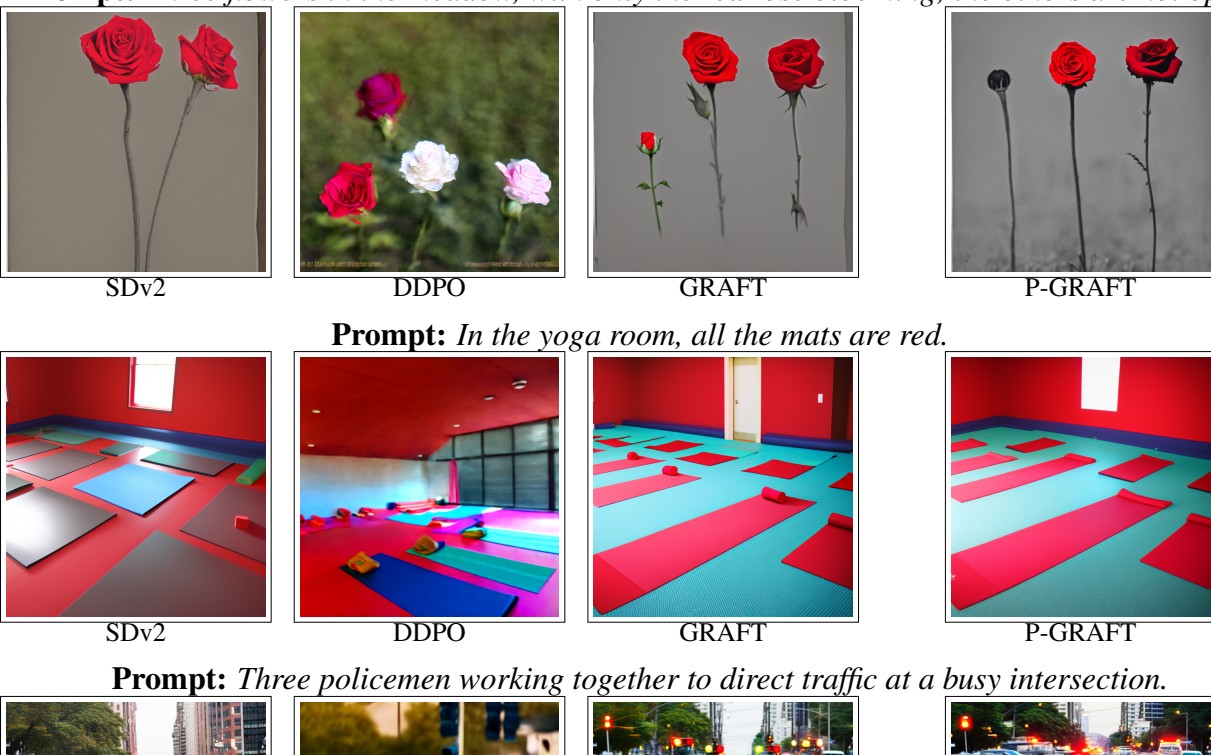

**Prompt:** *In the yoga room, all the mats are red.*

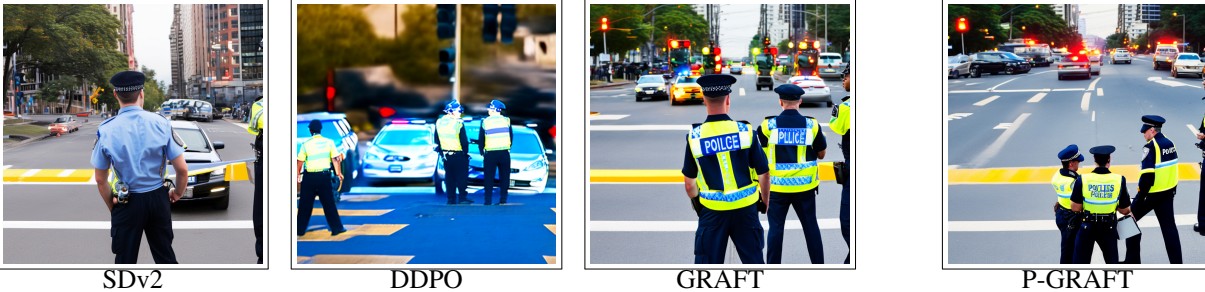

**Prompt:** *Three policemen working together to direct traffic at a busy intersection.*

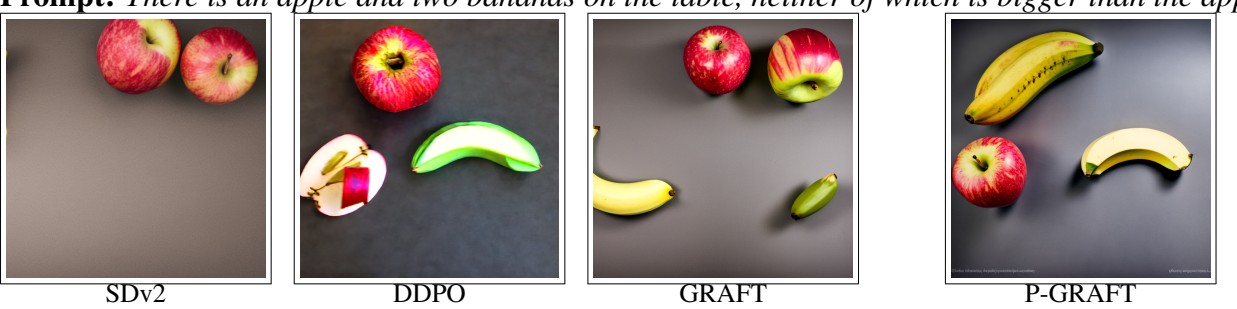

**Prompt:** *There is an apple and two bananas on the table, neither of which is bigger than the apple.*

Figure 5: Qualitative examples on GENAI-BENCH. All results are reported for the same seed across different algorithms.

### E.5.2 T2I-COMPBENCH++

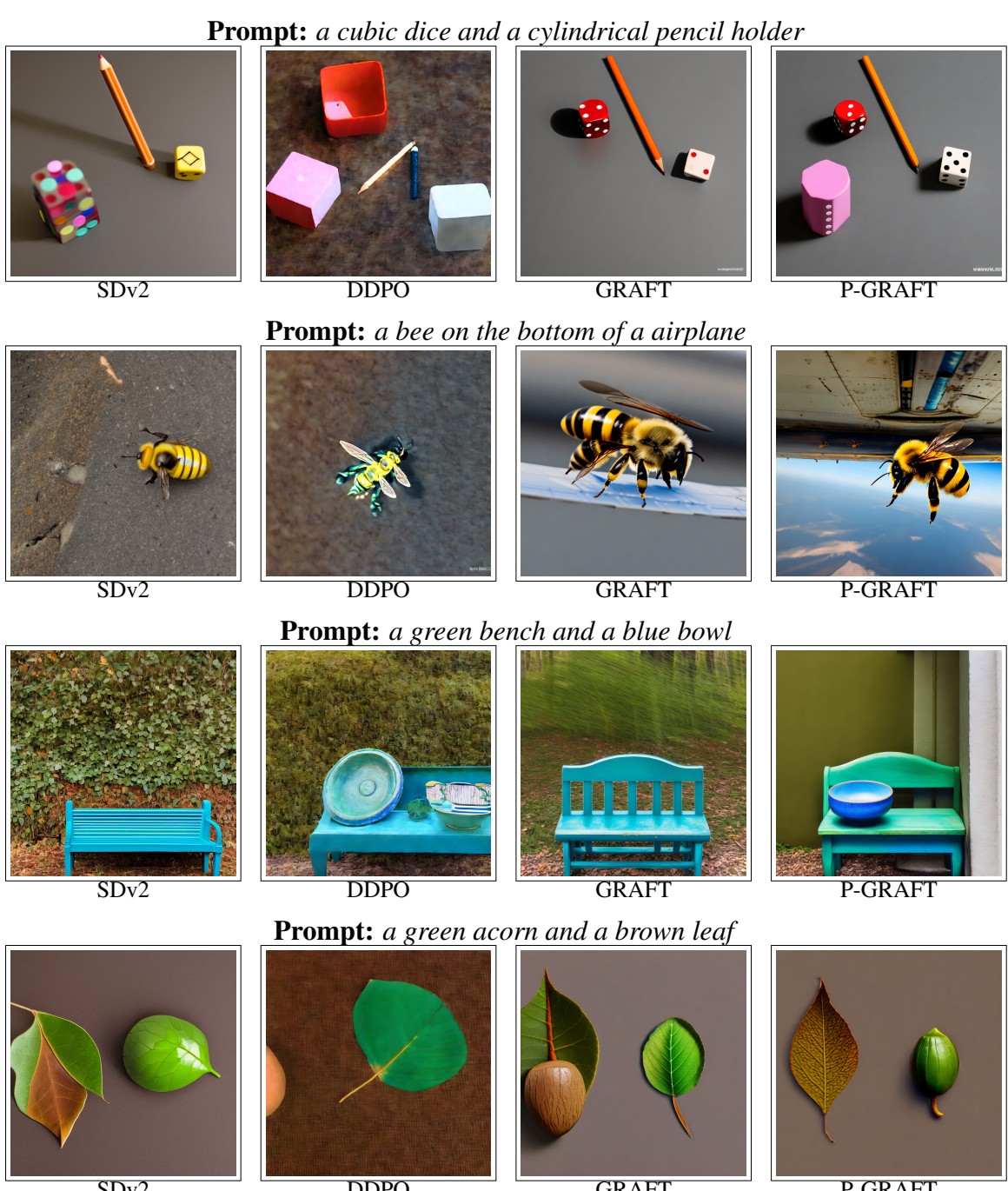

Figure 6: Qualitative examples on T2I-COMPBENCH. All results are reported for the same seed across different algorithms.

# F LAYOUT AND MOLECULE GENERATION

## F.1 INTERLEAVED GIBBS DIFFUSION (IGD)

Fine-tuning for both layout generation and molecule generation are done using models pre-trained using the Interleaved Gibbs Diffusion (IGD) (Anil et al., 2025) framework. IGD performs well for discrete-continuous generation tasks with strong constraints between variables - and hence is particularly useful for tasks like layout generation and molecule generation. Further, IGD offers a generalizable framework which can be used for both tasks - while other discrete-continuous diffusion frameworks exist, they are specialized to a particular task, often using domain specific adaptations.

On a high-level, IGD interleaves diffusion for discrete and continuous elements using Gibbs-style noising and denoising. Essentially, discrete elements are noised using flipping and trained using a binary classification loss. Continuous elements use typical DDPM-style noising and training. While the exact forward and reverse processes are different from DDPM-style processes which we have considered in the main text, the key results follow empirically and theoretically.

## F.2 LAYOUT GENERATION

**Problem Formulation:** A layout is defined as a set of $N$ elements $\{e_i\}_{i=1}^N$. Each element $e_i$ is represented by a discrete category $t_i \in \mathbb{N}$ and a continuous bounding box vector $\mathbf{p}_i \in \mathbb{R}^4$. Following (Anil et al., 2025), we use the parameterization $\mathbf{p}_i = [x_i, y_i, l_i, w_i]^\top$, where $(x_i, y_i)$ represents the upper-left corner of the bounding box, and $(l_i, w_i)$ its length and width, respectively. *Unconditional* generation represents generation with no explicit conditioning for the elements, whereas *Class-Conditional* generation indicates generations conditioned on element categories.

**Implementation Details:** For pre-training, we follow the exact strategy used in (Anil et al., 2025). Fine-tuning is also done with the same hyperparameters used for pre-training. Since the data and model sizes are significantly smaller compared to images, each round of rejection sampling is done on 32768 samples, of which the top $50\%$ samples are selected. For each sampling round, 10000 training iterations are performed with a training batch size of 4096. The results reported in Table 3 are for 20 such sampling rounds. FID computation is done by comparing against the test split of PubLayNet.

## F.3 MOLECULE GENERATION

**Problem Formulation:** The task of molecule generation involves synthesizing molecules given a dataset of molecules. A molecule consists of $n$ atoms denoted by $\{z_i, \mathbf{p}_i\}_{i=1}^n$, where $z_i \in \mathbb{N}$ is the atom's atomic number and $\mathbf{p}_i \in \mathbb{R}^3$ is the position. A diffusion model is trained to generate such molecules. In this work, we take such a pre-trained model, and try to increase the fraction of stable molecules, as deemed by RDKit.

**Implementation Details:** For pre-training, we follow the exact strategy used in (Anil et al., 2025). Fine-tuning is also done with the same hyperparameters used for pre-training. Since the data and model sizes are significantly smaller compared to images, each round of rejection sampling is done on 32768 samples. We select all stable molecules, but with the de-duplication strategy described in Section 2.3 - we find that *this is crucial* to maintain diversity of generated molecules. For each sampling round, 10000 training iterations are performed with a training batch size of 4096. The $1\times$ in Table 4 corresponds to 10 such sampling rounds - $9\times$ therefore corresponds to 90 sampling rounds.

**Uniqueness of Generated Molecules:** To demonstrate that the fine-tuned models still generate diverse molecules, and do not collapse to generating a few stable molecules, we report the uniqueness metric computed across the generated molecules below. From Table 20, it is clear that the fine-tuned models still generate diverse samples since the uniqueness of the generated molecules remain close to the pre-trained model. Uniqueness is as determined by RDKit.

**Effect of de-duplication** We also try out an ablation where we use GRAFT, but without the de-duplication - i.e., we train on all stable molecules irrespective of whether they are unique or not. The results are shown in Figure 7 - without de-duplication, it can be seen that though stability is recovered,

Table 20: Uniqueness of generated molecules

| Model | Mol: Stability | Uniqueness |
|---|---|---|
| Baseline | $90.50_{\pm0.15}$ | $95.60_{\pm0.10}$ |
| GRAFT | $90.76_{\pm0.20}$ | $96.04_{\pm0.46}$ |
| P-GRAFT($0.5N$) | $90.46_{\pm0.27}$ | $95.70_{\pm0.28}$ |
| P-GRAFT($0.25N$) | $\mathbf{92.61}_{\pm0.13}$ | $95.32_{\pm0.07}$ |

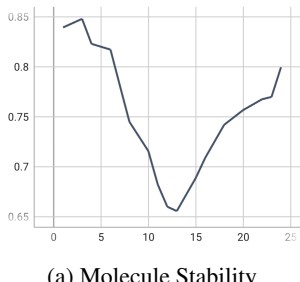

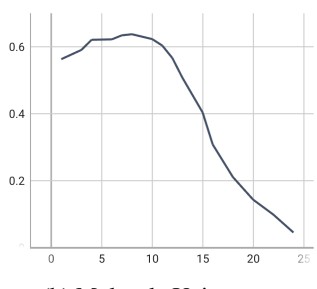

(a) Molecule Stability

(b) Molecule Uniqueness

Figure 7: Molecule Stability and Uniqueness without De-duplication

uniqueness is lost, indicating that the model produces only a small subset of molecules it was initially able to produce.

**Fine-Tuning without Predictor-Corrector:** IGD makes use of a version of predictor-corrector method (Lezama et al., 2022; Zhao et al., 2024; Campbell et al., 2022; Gat et al., 2024) termed ReDeNoise at inference-time to further improve generations. The results reported so far make use of this predictor-corrector. While ReDeNoise improves performance significantly, it comes at the cost of higher inference-time compute. We report results of the baseline and fine-tuned version without ReDeNoise in Table 21. Both GRAFT and P-GRAFT still show improvement over the baseline, even without ReDeNoise.

Table 21: Results for Molecule Generation without ReDeNoise

| Model | Mol: Stability | Sampling Steps |
|---|---|---|
| Baseline | 84.00 | - |
| GRAFT | 87.13 | $9\times$ |
| P-GRAFT ($0.5N$) | 84.57 | $1\times$ |
| P-GRAFT ($0.25N$) | **88.36** | $1\times$ |

# G    INVERSE NOISE CORRECTION

## G.1    IMPLEMENTATION DETAILS

The pre-trained models, corresponding to TRAIN_FLOW function in Algorithm 3, are trained using the NSCNpp architecture and hyperparameters from the official codebase of Song et al. (2020a) with minor changes which we describe below. The noise corrector model is also trained with the same architecture except that the number of channels are reduced from the original 128 to 64 channels - this leads to a reduction in parameter count by $\approx 4\times$. For the pre-trained model, we train with `num_scales` $= 2000$, positional embeddings and a batch size of 128. For the noise corrector model, we use the same hyperparameters except for `num_scales` $= 1000$. FID with 50000 samples is used to measure the performance, as is standard in the literature. Note that a separate noise corrector model is trained for each choice of $\eta$ in Algorithm 3, i.e., for the results reported in Table 5, separate noise corrector models are trained for pre-trained steps of 100 and 200.

**CelebA-HQ:** For the baseline pre-trained flow model, we use the checkpoint after 330k iterations, since this gave the lowest FID. For noise corrector model training, we use this checkpoint to generate the inverse noise dataset and train on it for 150k iterations.

**LSUN-Church:** For the baseline pre-trained flow model, we use the checkpoint after 350k iterations, since this gave the lowest FID. For noise corrector model training, we use this checkpoint to generate the inverse noise dataset and train on it for 55k iterations. Note that Backward Euler (Algorithm 6) suffered from numerical instability, which we hypothesize is due to plain backgrounds, when done on LSUN-Church. To alleviate this issue, we perturb the images with a small Gaussian noise $\mathcal{N}(0, \sigma^2 \mathbf{I})$, with $\sigma = 10^{-3}$.

## G.2    FLOPS COMPARISON

We present a comparison of the exact FLOPs used for inference:

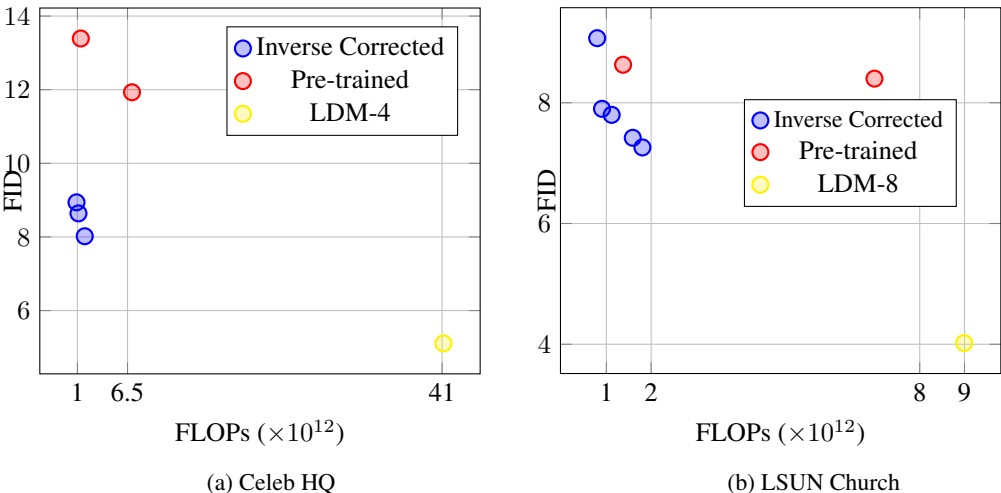

(a) Celeb HQ                    (b) LSUN Church

Figure 8: **FLOPs vs FID**: The inverse corrected model achieves better FID despite incurring lower FLOPs. Corresponding LDM models have been added for both datasets for reference.

