# OpenReview forum: "Fine-Tuning Diffusion Models via Intermediate Distribution Shaping"
_ICLR.cc/2026/Conference — ICLR 2026 Poster_

### Official Review · Reviewer_xSnM · 2025-10-28

**Soundness:** 2
**Presentation:** 3
**Contribution:** 2
**Rating:** 6
**Confidence:** 3

**Summary:**

This paper introduces fine-tuning methods for diffusion models through intermediate distribution shaping. The authors first unify rejection sampling-based fine-tuning methods as GRAFT (Generalized Rejection sAmpling Fine-Tuning) and prove it implicitly performs PPO with reshaped rewards, enabling marginal KL regularization for diffusion models despite intractable likelihoods. They then propose P-GRAFT (Partial-GRAFT), which fine-tunes only until intermediate denoising timesteps by assigning final generation rewards to partial noisy states. This is justified through a bias-variance tradeoff: while variance of rewards conditioned on intermediate states increases with noise level (Lemma 4.3), the score function becomes easier to learn at higher noise levels (Theorem 4.4). Additionally, they introduce Inverse Noise Correction for flow models, which trains a model to correct the distribution shift in initial noise without explicit rewards. Empirically, P-GRAFT applied to Stable Diffusion v2 achieves 8.81% relative improvement in VQAScore over the base model on text-to-image benchmarks and outperforms policy gradient methods. For unconditional generation, Inverse Noise Correction improves FID at lower FLOPs/image.

**Strengths:**

* The unification of rejection sampling methods (RAFT, RSO, BoN) under the GRAFT framework with explicit connection to PPO is novel and theoretically grounded (Lemma 3.2)
* P-GRAFT represents a genuinely new approach to diffusion model fine-tuning by operating at intermediate noise levels rather than final outputs
* The bias-variance analysis providing theoretical justification for intermediate distribution shaping is insightful and well-executed
* Mathematical formulations are generally clear with good notation

**Weaknesses:**

1. **Limited theoretical analysis for P-GRAFT optimality**: There is no formal analysis of the optimal choice of $N_I$ or convergence guarantees. The paper relies heavily on empirical validation across different $N_I$ values, but theoretical guidance for selecting $N_I$ would strengthen the contribution.

2. **Assumption dependencies**: Lemma 5.1 requires $\eta L < 1$ for backward Euler, but the paper doesn't discuss how this constraint affects practical step size choices or what happens when the Lipschitz constant $L$ is large. The impact on computational cost is not analyzed.

3. **Gap between theory and practice**: The theoretical analysis assumes continuous-time SDEs, but experiments use discrete-time DDPM/DDIM schedulers. The discretization error and its interaction with P-GRAFT is not discussed.

**Questions:**

1. How does the optimal $N_I$ depend on task characteristics (reward smoothness, data modality, model capacity)? Can you provide theoretical or empirical guidance for selecting $N_I$?

2. Theorem 4.4 shows score functions at later times are exponentially closer to Gaussian. Have you verified this empirically by measuring $H_t^T$ at different timesteps?

3. The backward Euler method (Algorithm 6) requires solving a fixed-point equation. How many iterations $N_b$ are typically needed in practice, and what is the computational cost relative to forward sampling?

4. Can P-GRAFT be combined with classifier-free guidance training? How does the guidance scale interact with the intermediate timestep choice?

---

> ### Author Response · Authors · 2025-11-19
> **Regarding weaknesses**
>
> We thank the reviewer for pointing out that P-GRAFT is novel, the bias-variance analysis is insightful and that mathematical formulations are clear. We now address some concerns raised by the reviewer.
>
> > Limited theoretical analysis for P-GRAFT optimality: There is no formal analysis of the optimal choice of $N_I$ or convergence guarantees. The paper relies heavily on empirical validation across different $N_I$ values, but theoretical guidance for selecting $N_I$ would strengthen the contribution.
>
>
> Thank you for raising this question. While we have tried analyzing the choice of $N_I$ formally, we could not arrive at a mathematical method for estimating $N_I$. This can be highly dependent on the dataset in question as well as the reward function. We believe deriving the best $N_I$ in general is a highly non-trivial problem and beyond the scope of our current work.  Given the fact that $N_I$ is a hyper-parameter, we also note that finding the optimal choice of hyper-parameters for neural network algorithms is a hard and long standing open problem in this field.
>
> However, heuristics can help find potential values for $N_I$ through inference rollouts. Typically, following the bias-variance tradeoff discussed in Section 4, $N_I$ with a relatively low expected conditional variance would be a good choice for P-GRAFT. We have reported these values for molecule generation (Table 1) and text-to-image generation (Table 10).
>
> > Assumption dependencies: Lemma 5.1 requires $\eta L < 1$ for backward Euler, but the paper doesn't discuss how this constraint affects practical step size choices or what happens when the Lipschitz constant $L$ is large. The impact on computational cost is not analyzed.
>
> The values of $\eta$ and $L$ determine when the forward Euler iteration can be inverted. In our experiments, for the step sizes considered, we were able to invert each forward Euler iteration with num_steps = 100 (step size = 1/100). However, the inversion was not possible when num_steps <100. In fact, this choice of step-size is well known in the numerical simulation of ODEs. Specifically, to run Euler discretization for an ODE with $L$-Lipschitz flow, the step size should be chosen such that $\eta<1/L$.
>
> > Gap between theory and practice: The theoretical analysis assumes continuous-time SDEs, but experiments use discrete-time DDPM/DDIM schedulers. The discretization error and its interaction with P-GRAFT is not discussed.
>
> It suffices to understand just the continuous-time SDEs  in our case. This can be seen by first analyzing the sources of error in diffusion model sampling (learning error + discretization error) and then looking at how to bound the statistical error using continuous-time SDEs.
>
> **Causes for Error:**
>
> As per theoretical results in the literature [1], the final generation error can be decomposed into:
>
> **(a)** imperfect learning during training and **(b)** discretization
>
> (a) depends on how well we learn the score function across different times and (b) depends only on how coarse our discretization is.
>
> Since we consider all our algorithms to have the same discretization schedule, it is sufficient to understand how well the training algorithm learns the score function in order to understand which algorithms perform better.
>
> **Error in Learning Score Function:**
>
> Our work shows that it is easy to learn the score function, so that (a) discussed above can be bounded. The standard loss formulation for training via score matching is an integral across time (See Equation (17) in [2] and Equation (7) in [3]). Here, $x_t$ is drawn from the forward process given $x_0$ (the actual image data point). The forward process has a closed form solution, and $x_t$ can be obtained in closed form without discretized schedulers like DDIM/ DDPM. The standard diffusion model training algorithms sample $t$ from a continuous distribution, without discretization and obtain $x_t$ given $x_0$ using closed form solution to the forward process. We refer to the standard codebase in this area, corresponding to [2] (user:yang-song, project:score_sde, file: losses in GitHub).  The 'time' during training is sampled uniformly from a continuous distribution when training.continuous = True and $x_t$ is sampled from the forward process given $t$.
>
>
> Therefore, **the training is independent of the choice of discretization** (which is only relevant during inference). Thus, we believe that our analysis considering the forward SDE is sufficient.
>
> **References**
>
> [1] Nearly d-Linear Convergence Bounds for Diffusion Models via Stochastic Localization, Benton et al, arXiv:2308.03686
>
> [2]: Denoising Diffusion Probabilistic Models, Ho et al, arXiv: 2006.11239
>
> [3]: Score-Based Generative Modeling Through Stochastic Differential Equations, Sohl-Dickstein et al, arXiv:2011.13456

---

> ### Author Response · Authors · 2025-11-19
> **Regarding questions**
>
> > Theorem 4.4 shows score functions at later times are exponentially closer to Gaussian. Have you verified this empirically by measuring $H_t^T$ at different timesteps?
>
> We have not verified this empirically. However, this is guaranteed to hold as per the well-established Bakry-Emery Theory of Diffusion Processes as mentioned in the manuscript.
>
> > The backward Euler method (Algorithm 6) requires solving a fixed-point equation. How many iterations $N_b$ are typically needed in practice, and what is the computational cost relative to forward sampling?
>
> Since the backward update is contractive in nature, the rate of convergence to the previous point $x_{t-1}$ is exponential at each time step $t$. For all our experiments, we observed that $N_b=10$ (in many cases even less) sufficed. We have mentioned these details for the ODE Solver Algorithms in Appendix B.
>
> The reviewer is correct regarding the difference between the computational cost of the forward and backward sampling. Roughly speaking, the backward computation is $N_b$ times more than the forward update. However, we would like to point out that the backward computation **only needs to be done once** when we want to generate the inverse noise dataset for training the noise corrector model. During real time inference, we **only run the forward sampler** and as demonstrated through our empirical results, inverse noise correction can lead to **significantly fewer FLOPs** and faster generation.
>
> > Can P-GRAFT be combined with classifier-free guidance training? How does the guidance scale interact with the intermediate timestep choice?
>
> We use P-GRAFT with classifier-free guidance for all our experiments, **keeping the guidance scale the same** - we have mentioned the exact guidance scale values in the Appendix. The exact interaction between P-GRAFT and classifier-free guidance has additional conceptual nuances and is beyond the scope of current work. Some nuances to consider are:
>
> - Lower guidance scales usually lead to higher diversity but lower quality/ alignment. This can be helpful while performing rejection sampling to obtain diverse samples which can prevent mode collapse.
>
> - Higher guidance scales usually lead to better quality/alignment and can thus provide high-quality samples to train on.
>
> - Using the same guidance scale during rejection sampling and training can potentially prevent distribution shifts during inference.

---

> ### Comment · Reviewer_xSnM · 2025-11-23
>
> I appreciate the author's response and analysis regarding design selection, which made the paper clearer and address some of my concerns. Thus I increase the soundness score to 3.  Overall, I think the problems this paper aims to address, such as the fine-tuning of intermediate time and sampling methods in RL, are meaningful. However, I believe the paper could benefit from further addressing and discussing the following concerns and minor issues.
>
> (a) "Given the intractability of (marginal)PPO, we explore conceptual connections between rejection sampling based fine-tuning methods and PPO." Could you provide a clearer explanation of this sentence? This could help to better understand the motivation of this paper and avoid misunderstandings. In addition, perhaps there should be a space around "(marginal)" (minors).
>
> (b) Reducing the number of FLOPs operations is interesting, but it would be even better if metrics related to computation time could be provided (e.g., training/inference).
>
> (c) Furthermore, since stable diffusion 2, SDv2, and Stable Diffusion v2 appear throughout the text, I recommend to use them consistently to avoid misunderstandings, or at least to explain their meaning the first time SDv2 appears to avoid misunderstandings.
>
> (d) In Top − K Sampling, why is there a 0 in the analysis formula for $\frac{\hat{r}(x)}{\alpha}$? Could you explain this part in more detail?
>
> (e)  I'd like to know what advantages and significant differences this method has compared to the current best-of-N, Reward-Anked FineTuning (RAFT) method (this could help readers better understand the core contributions of the paper)? Could you further elaborate on the difference between P-GRAFT and GRAFT?

---

> > ### Author Response · Authors · 2025-11-24
> > **Further clarifications**
> >
> > We thank the reviewer for raising the soundness score. We now address the concerns and minor issues raised by the reviewer:
> >
> > > (a) "Given the intractability of (marginal)PPO, we explore conceptual connections between rejection sampling based fine-tuning methods and PPO." Could you provide a clearer explanation of this sentence? This could help to better understand the motivation of this paper and avoid misunderstandings. In addition, perhaps there should be a space around "(marginal)" (minors).
> >
> >
> > We have added a more detailed explanation in the introduction of the revised manuscript (Lines 37-44). We reproduce it below for ease of access:
> >
> > Mathematically, suppose $p(\cdot)$ is the distribution induced by a generative model. Given a sample (not necessarily generated by the model), we can compute the likelihood of $x = (x^1, \dots, x^n)$ for AR models as $p(x^1)p(x^2|x^1)...p(x^n|x^1, \dots, x^{n-1})$. This is because AR models predict the next token distribution conditioned on the previous tokens. This likelihood can directly be used for RL algorithms. Diffusion models, on the other hand, model the conditional likelihoods along the denoising time axis, i.e. we only have access to $p(x_{t-1}|x_{t})$, the conditional distribution of the denoised state given the previous state. From these conditionals, it is intractable to compute the marginal likelihood $p(x)$. This makes the marginal likelihood unavailable to implement popular RL algorithms like PPO.
> >
> >
> > > (b) Reducing the number of FLOPs operations is interesting, but it would be even better if metrics related to computation time could be provided (e.g., training/inference).
> >
> > While re-running the training to report computation time is difficult due to computational constraints, we have added computation time for inference to Table 5 of the revised manuscript. The trend in FLOPs carries over directly; Inverse Noise Correction gives quality improvement at reduced inference time. We reproduce the numbers here for ease of access:
> >
> > |Noise Corrector (Steps) | Pre-Trained Model (Steps) | Inference Time (Seconds)|
> > |--------------------------|------------------------------|--------------------------------|
> > | - | 1000 | 482.85|
> > | - | 200 | 96.64 |
> > |100 | 100 | 67.85 |
> > | 200 | 200 | 135.42 |
> >
> >
> > > (c) Furthermore, since stable diffusion 2, SDv2, and Stable Diffusion v2 appear throughout the text, I recommend to use them consistently to avoid misunderstandings, or at least to explain their meaning the first time SDv2 appears to avoid misunderstandings.
> >
> > We thank the reviewer for pointing this out. We have changed the text to use solely “Stable Diffusion v2” and “SDv2”. Further, we have indicated that SDv2 refers to Stable Diffusion v2 in Line 402 of the revised manuscript.
> >
> > > (d) In Top − K Sampling, why is there a 0 in the analysis formula for $ \frac{\hat{r}(x)}{\alpha} $? Could you explain this part in more detail?
> >
> > Thank you very much for pointing this out; we believe the reviewer is referring to the equation in Line 134 of the revised manuscript. This was a typo; there should not be a 0 in the equation. We have corrected this in the revised manuscript.

---

> > ### Author Response · Authors · 2025-11-24
> > **Further clarifications (ctd.)**
> >
> > >  (e) I'd like to know what advantages and significant differences this method has compared to the current best-of-N, Reward-Anked FineTuning (RAFT) method (this could help readers better understand the core contributions of the paper)? Could you further elaborate on the difference between P-GRAFT and GRAFT?
> >
> > We have added additional explanations in Lines 97-102 and Lines 179-187 of the revised manuscript. We summarize it again below for ease of access:
> >
> > GRAFT conceptually unifies current rejection sampling methods such as Best-of-N (BoN) and RAFT. Further, going beyond BoN and Top-K (as adopted by RAFT variants), our generalization admits novel rejection sampling strategies such as those involving de-duplication. Note that we have explicitly acknowledged the special cases in the manuscript ( and hence the name GRAFT). We establish theoretically that GRAFT is particularly useful for diffusion because it is a simple strategy which allows implicit sampling from the solution to the optimization objective of PPO. Empirically, we show that GRAFT works well for large-scale prompt sets and outperforms policy-gradient methods. GRAFT is a unifying formalism to motivate our novel algorithmic contribution, which is P-GRAFT.
> >
> > In P-GRAFT, the distribution shaping is done at an intermediate step rather than at the final denoised output as is done in GRAFT. Intuitively, the higher noise levels decide the "outline" of the image to be denoised and is sensitive to the conditioning. Additionally, fine-tuning denoisers at lower noise levels require relatively more samples because learning is more difficult. Thus, we introduce P-GRAFT to fine-tune only the decision-making higher noise levels. For the rest of the denoising process, it suffices to revert to the original denoiser. Theoretically, we show that although this results in higher reward variance (due to reward being used to select a partially noisy state), learning becomes easier due to lower bias at higher noise levels. Empirically, this results in consistent performance improvements over GRAFT (and other baselines) for text-to-image benchmarks with Stable Diffusion v2. We would like to emphasize that these bias-variance nuances have not been exploited in RL for diffusion literature as far as we are aware.

---

### Official Review · Reviewer_sMVE · 2025-10-28

**Soundness:** 2
**Presentation:** 2
**Contribution:** 2
**Rating:** 2
**Confidence:** 5

**Summary:**

The authors investigate:
(1) Rejection sampling and it’s uses in sampling and fine-tuning diffusion models. The authors derive the sampling distribution for top-k sampling, with arbitrary reward functions and de-duplication criterion, for diffusion models. The authors then propose fine-tuning on the trajectories yielded by rejection sampling.

(2) An inverse noise correction algorithm that trains a flow model from N(0, 1) to the t=1 distribution of the flow model, and then sample from the flow model.

**Strengths:**

See questions

**Weaknesses:**

See questions

**Questions:**

Will the authors clarify what their contributions are exactly, as

1. The sampling distribution of top-k sampling is known:
    1. As the authors acknowledge, the derivations are known from Amini et al 2024. The derivations there are not specific to any particular kind of method of generation, it applies to any generative model, including both AR, diffusion models. See theorem 2 in Amini et al 2024.
2. Top-k sampling for fine-tuning has been proposed and shown in RAFT, see Dong et al

Questions regarding fine-tuning with Top-k sampling:

1. What distribution does alg 2 (P-GRAFT inference) sample?  Alg 2 stiches together two different diffusion models, with the first one being fine-tuned and sampling from the reward tilted intermediate distribution, followed by the base model.

Questions regarding the motivation for inverse-noise correction:

1. The motivation for the method is not clear from the text, either in the main paper or appendix.

Experimental Clarifications:

1. Can the authors clarify what objective was used to fine-tune the diffusion model in alg 1 (P-GRAFT train)
2. The GenEval scores reported in the paper for SDv2 and SDXL-base are higher than those reported in the GenEval paper. Can the authors clarify what prompts were used for producing the numbers reported in table 2.

Minor clarification regarding using the term PPO distribution: PPO is a method for learning the reward tilted distribution, p(x) exp(r(x)). This distribution can be learned using methods other than PPO.

---

> ### Author Response · Authors · 2025-11-19
> **Regarding contributions**
>
> We thank the reviewer for reviewing our work. We now address the concerns raised by the reviewer.
>
> > Will the authors clarify what their contributions are exactly
>
> We first note that GRAFT is **only one of our three contributions** - we also introduce P-GRAFT (a novel algorithmic improvement over GRAFT) as well as Inverse Noise Correction (a novel error correction strategy for flow models) in Sections 4 and 5 respectively. In fact, as evidenced by the title, our main point is that titling at an intermediate time-step can be more effective for diffusion (and flow) models.
>
> GRAFT is an introductory formalism that unifies rejection sampling methods and connects them to reward-shaped PPO objectives. Further, we empirically demonstrate the effectiveness of this simple framework.
>
> > As the authors acknowledge, the derivations are known from Amini et al 2024. The derivations there are not specific to any particular kind of method of generation, it applies to any generative model, including both AR, diffusion models. See theorem 2 in Amini et al 2024.
>
> Note that Amini et al 2024 considers only **Best-of-N distribution**, and not the Top-K sampling distribution. Notwithstanding this, Generalized Rejection Sampling (GRS) as introduced in Definition 3.1 is **more general**. For instance, GRS also subsumes rejection sampling schemes for binary rewards where all high reward samples are selected with de-duplication. This does not come under top-K sampling, but does come under GRS. In fact, we specifically use this scheme for molecule generation. Again, this is only a formalism to unify the connection between rejection sampling and PPO style objectives before moving onto other results surrounding P-GRAFT.
>
> > Top-K sampling for fine-tuning has been proposed and shown in RAFT, see Dong et al
>
> Note that we have acknowledged Dong et al in the manuscript and in fact named the algorithm GRAFT following their nomenclature, i.e., RAFT. As discussed above, while they propose Top-K sampling based fine-tuning, the specific utility of this algorithm in the context of diffusion, namely **enabling implicit marginal KL constraint, is not discussed**. More importantly, they do not report any improvement over DDPO in terms of CLIP score - however, for large prompt sets, we **empirically demonstrate gains across datasets and tasks**. We also develop P-GRAFT,  a novel algorithm, for further gains.

---

> ### Author Response · Authors · 2025-11-19
> **Regarding P-GRAFT distribution**
>
> > What distribution does alg 2 (P-GRAFT inference) sample? Alg 2 stiches together two different diffusion models, with the first one being fine-tuned and sampling from the reward tilted intermediate distribution, followed by the base model.
>
> This is straightforward to compute using Lemma 4.2. We have added the computation in Appendix D.3.1 in the revised manuscript. If the reference diffusion policy is $\bar{p}$,  the stitched distribution $p^s$ looks like:
>
> $$  p^s(x_0) \propto \int \bar{p}_{0|t}(x_0 | x_t) \bar{p}_t(x_t)  \exp{ (\frac{ \hat{r}(x_t)} {\alpha})}  dx_t   $$
>
> In general, this integral is intractable. However, we have added two edge cases in the discussion to provide an intuitive understanding of the resulting tilt. We provide the high-level intuition behind this discussion here:
>
> - Consider the case where the intermediate latent at time $t$, i.e., $x_t$ is independent of the reward obtained after complete denoising. That is,  $x_t$ has no information about the eventual reward that is obtained. This would naturally translate to high conditional reward variance. In this case, we show that $p^s$ is identical to the original reference distribution $\bar{p}$, i.e., **there is no tilt whatsoever**.
> - On the other hand, consider the case where the reward obtained after complete denoising is completely determined by $x_t$, i.e. given $x_t$, the reward obtained after denoising is deterministic. This would correspond to zero conditional reward variance since the reward is fixed under conditioning. We show that in this case $p^s$ is **identical to the distribution induced by GRAFT**, i.e. the solution to the PPO objective.
>
> For intermediate cases, therefore, P-GRAFT interpolates between the two distributions described above. This is precisely what we quantify using conditional variance through Lemma 4.3.

---

> ### Author Response · Authors · 2025-11-19
> **Regarding motivation for Inverse Noise Correction and experimental clarifications**
>
> > The motivation for the method is not clear from the text, either in the main paper or appendix.
>
> As discussed throughout Section 5, and specifically in Lines 372-377, we use inverse noise correction to correct for the errors in the learned distribution by learning the "correct" noise distribution which is not Gaussian. This is also made clear through Figure 2 and Algorithm 3.
>
> Could the reviewer clarify which part of the explanation in Section 5 is not clear as we have explicitly stated the types of errors and how Inverse Noise Correction mitigates it?
>
> > Can the authors clarify what objective was used to fine-tune the diffusion model in alg 1 (P-GRAFT train)
>
> As discussed in the manuscript, we use the pre-training algorithm for fine-tuning. We have given more details in the Appendix (Lines 1601 - 1638 in the revised manuscript).
>
> > The GenEval scores reported in the paper for SDv2 and SDXL-base are higher than those reported in the GenEval paper. Can the authors clarify what prompts were used for producing the numbers reported in table 2.
>
> As mentioned in Lines 403 and 432, what we report is **VQAScore on the GenEval dataset**, whereas the GenEval paper reports GENEVAL score on the GenEval dataset. We use VQAScore since it is more recent and refined [1]. We use the complete GenEval prompt set, as is standard.
>
> **References**
>
> [1]  Evaluating Text-to-Visual Generation with Image-to-Text Generation, arxiv:2404.01291

---

### Official Review · Reviewer_iasd · 2025-10-30

**Soundness:** 3
**Presentation:** 1
**Contribution:** 2
**Rating:** 2
**Confidence:** 3

**Summary:**

The paper recognizes the importance of reward-guided fine-tuning of diffusion models and the complexity of leveraging proximal PG methods due to intractable likelihoods. The authors introduce a unified rejection sampling framework, show that fine-tuning according to it leads to the same solution of proximal PG schemes, and introduce a method for controlling also the tilt of intermediate distributions of the diffusion process. They present a mathematical analysis for bias-variance trade-off, and propose a noise-correction scheme for reward-independent improvement of flow models. Ultimately, the present an experimental evaluation of the proposed contributions.

**Strengths:**

- The paper correctly identifies a limitation of RL-based schemes for fine-tuning of diffusion models (i.e., intractable log likelihoods), a timely task with high-relevance.

- The idea proposed within Sec. 5 seems interesting, but I could not fully understand its logic on an intuitive level.

- The experimental sections includes diverse dataset spanning images, layout gen., and molecular design.

**Weaknesses:**

The paper presents an excessive amount of ideas, each not sufficiently justified, motivated, or explained. As the paper structure is very complex due to the multitude contributions, I will discuss weaknesses of each contribution a-d as listed in Sec. 1.

**a)**
1. The paper often seems to confuse a *problem*, in particular solving an entropy-regularized MDP, with an algorithm, i.e. PPO. I understand that the presented algorithm has solution corresponding to the optimal solution of entropy-regularized MDPs, which can be tackled by PPO, but what does 'GRAFT enables PPO' or 'implicitly perform PPO' even mean? This confusion seems repeated several times within the work. Concretely, I believe these statements are effectively wrong, as the propose method, while inducing the same solution, does *not* enable/perform PPO.
2. It is well-known that a general class of inference-time schemes for diffusion induces this solution class (see [1], Eq. 1). One such case is [2] (see Sec. 3.2). So it seems there is already a variety of inference-time schemes solving this problem via diverse techniques (c.f., [1]). Moreover, there exist fine-tuning control-theoretic schemes that can solve this entropy-reg. problem as well (e.g., [3]) without value bias problem, as also mentioned by the authors. Ultimately, leveraging inference-time schemes solving this problem to then fine-tune a model, which seems to me the core algorithmic idea here, is already presented in [1, Sec. 9] arguably in a more performative fashion than the one presented within this paper (i.e. , via policy distillation rather than plain training). As a consequence, it seems to me that there isn't significant novelty within the presented contribution.

**b)**
1. The KL is typically enforced on the data level distribution as a way to preserve the high-probability set learned by the pre-trained model. The justifications presented within the paper (both in theory 4.1 and experiments) do not seem convincing to me regarding why we should instead enforce the KL-reg. at another time-step. In particular, it seems to me that the theoretical investigation in Sec. 4.1 does not really provide a concrete answer to this question. In a sense, it shows that this problem might be easier, but this does not imply (practical or theoretical) relevance. Since the authors here are presenting a novel problem setting, it would be essential to motivate it clearly.

**c)**
- (writing) this section (i.e. Sec. 5) is not clearly written to the point that I could not fully grasp the presented idea. The problem tackled seems not particularly related with the problems treated in the rest of the paper, and is not sufficiently formalized to properly understand the gains of the proposed methodology. I would strongly suggest to also introduce algorithmic aspects with an intuitive presentation of their workings before/after presenting their implementation.


**Overall (writing/structure)**
The paper is poorly structured and lacks a solid narrative. Concretely, it presents multiple ideas without sufficient clarification of their motivation, and/or workings. The text often lacks conceptual explanations of new concepts/mechanisms and their implications.


**References**:

[1] Inference-Time Alignment in Diffusion Models with Reward-Guided Generation: Tutorial and Review, 2025

[2] Derivative-Free Guidance in Continuous and Discrete Diffusion Models with Soft Value-Based Decoding, 2024

[3] Adjoint Matching: Fine-tuning Flow and Diffusion Generative Models with Memoryless Stochastic Optimal Control, 2024

**Questions:**

- Did I misinterpret or misunderstand any of my points above within (a) or (b)?
- What is the core algorithmic intuitive idea for the method introduced in Sec. 5?

---

> ### Author Response · Authors · 2025-11-19
> **Regarding phrasing connecting GRAFT and PPO objective**
>
> We thank the reviewer for reviewing our work. We now address the concerns raised by the reviewer.
>
> > The paper often seems to confuse a problem, in particular solving an entropy-regularized MDP, with an algorithm, i.e. PPO. I understand that the presented algorithm has solution corresponding to the optimal solution of entropy-regularized MDPs, which can be tackled by PPO, but what does 'GRAFT enables PPO' or 'implicitly perform PPO' even mean? This confusion seems repeated several times within the work. Concretely, I believe these statements are effectively wrong, as the propose method, while inducing the same solution, does not enable/perform PPO.
>
> We thank the reviewer for pointing out this potentially confusing phrasing. As the reviewer has rightly pointed out, GRAFT induces the same solution in the policy space as the optimal solution to the PPO objective. While we have explicitly stated this (Line 118, 120 in the revised manuscript) as well as motivated the exact problem we are considering in the Preliminaries (Lines 79 - 87), we understand that using the phrase "GRAFT implicitly performs PPO" or "GRAFT enables PPO" can be confusing. What we meant by this is that GRAFT induces the same solution as the PPO objective - we have revised the manuscript with this change in phrasing.

---

> ### Author Response · Authors · 2025-11-19
> **Regarding Inference-Time schemes and Control-Theoretic schemes**
>
> > It is well-known that a general class of inference-time schemes for diffusion induces this solution class (see [1], Eq. 1). One such case is [2] (see Sec. 3.2). So it seems there is already a variety of inference-time schemes solving this problem via diverse techniques (c.f., [1]).
>
> As is clear from Eq 1 in [1] that reviewer has cited, these inference-time schemes all rely on **value functions**. Consequently, it does not subsume rejection sampling methods we consider, since these do not rely on value functions. Further, value function estimation requires computationally heavy methods such as Monte-Carlo Regression or first-order approximation methods such as posterior mean approximation (which introduces bias). We focus on rejection sampling methods precisely to avoid these complexities - GRAFT only requires filtered samples ($x_0$) and does not require value function estimation.
>
> GRAFT provides a common framework to analyze existing rejection sampling methods (such as Top-K sampling and RAFT) as well as the de-duplication based rejection sampling scheme introduced for molecule generation. Lemma 3.2 was stated for completeness - to establish that all these methods sample from the entropy-regularized MDP (with reward shaping). Further, the conceptual framework was introduced to set up the background necessary to introduce P-GRAFT.
>
> The main contribution, with respect to GRAFT, is **strong large-scale empirical performance across tasks**. We establish that simple rejection sampling based fine-tuning methods can outperform policy gradient methods for diffusion models. We believe this is a valuable contribution since policy gradient methods result in unstable training for diffusion.
>
> > Moreover, there exist fine-tuning control-theoretic schemes that can solve this entropy-reg. problem as well (e.g., [3]) without value bias problem, as also mentioned by the authors. Ultimately, leveraging inference-time schemes solving this problem to then fine-tune a model, which seems to me the core algorithmic idea here, is already presented in [1, Sec. 9] arguably in a more performative fashion than the one presented within this paper (i.e. , via policy distillation rather than plain training). As a consequence, it seems to me that there isn't significant novelty within the presented contribution.
>
> **Regarding policy-distillation of inference-time algorithms:**
>
> Following the reference cited by the reviewer, note that the master formulation in Section 9.2 requires explicit access to $p_{t-1}^* (\cdot | x_t) $. This requires computing value functions. As discussed above, this either requires computationally complex methods or first-order approximations.
>
> Further, the policy-distillation methods, as mentioned in the same reference, are sensitive to the choices of roll-in distribution and f-divergence (Section 9.2.1 and 9.2.2) - as opposed to this, GRAFT requires minimal hyper-parameter tuning and directly employs the pre-training objective. Finally, while many effective inference-time algorithms have been proposed, we are not aware of any works which demonstrate **empirical effectiveness of policy-distilled versions** of these algorithms, whereas we demonstrate effectiveness of GRAFT empirically.
>
> **Regarding control-theoretic schemes:**
>
> Note that we have discussed the limitations of these schemes in Appendix A. While it is true that [3] overcomes the value function bias problem, this framework is applicable only for ODE based methods such as rectified flow and DDIM. Hence this is more restrictive than our method, which is applicable for SDE methods as well. Similarly, while [4]  proposes a method that can work for SDEs as well, it is computationally expensive and requires tricks like gradient checkpointing.
>
>
> **References:**
>
> [1] Inference-Time Alignment in Diffusion Models with Reward-Guided Generation: Tutorial and Review, 2025
>
> [2] Derivative-Free Guidance in Continuous and Discrete Diffusion Models with Soft Value-Based Decoding, 2024
>
> [3] Adjoint Matching: Fine-tuning Flow and Diffusion Generative Models with Memoryless Stochastic Optimal Control, 2024
>
> [4] Fine-Tuning of Continuous-Time Diffusion Models as Entropy-Regularized Control, 2024

---

> ### Author Response · Authors · 2025-11-19
> **Regarding KL regularization at intermediate time-step**
>
> > The KL is typically enforced on the data level distribution as a way to preserve the high-probability set learned by the pre-trained model. The justifications presented within the paper (both in theory 4.1 and experiments) do not seem convincing to me regarding why we should instead enforce the KL-reg. at another time-step. In particular, it seems to me that the theoretical investigation in Sec. 4.1 does not really provide a concrete answer to this question. In a sense, it shows that this problem might be easier, but this does not imply (practical or theoretical) relevance. Since the authors here are presenting a novel problem setting, it would be essential to motivate it clearly.
>
> Thank you for bringing up this subtle but important point. We describe below how our theory and empirics support P-GRAFT.
>
> We first clarify that P-GRAFT does not present a novel problem setting - its goal is reward maximization of the final generated samples.
>
> To understand the KL regularization at an intermediate-step, first note that we use a **stitched model** for inference under PGRAFT - i.e., we use the fine-tuned model till the intermediate time-step, after which we switch to the reference model, as explicitly mentioned in Lines 190-191 as well as Algorithm 2.
>
> Intuitively, KL regularization at an intermediate time-step ensures that the generated latents at this time-step remain close (in distribution) to the latents of the reference model. Since the rest of denoising happens with the reference model, this also ensures that the **distribution of final generated samples remains close** to the sample distribution of the reference model.
>
> More formally, suppose the fine-tuned model obeys $KL(p_t || \bar{p}_t) \leq \alpha$, where $p_t$ is the fine-tuned model and $\bar{p}_t$ is the reference model. Let the final distribution induced by the stitched model be $p^s_0$.   Due to the contraction of KL divergences under Markov Chains, a.k.a **Data Processing Inequality** (Theorem 7.4 in [1]), $KL(p^s_0||\bar{p}_0) \leq \alpha$ because we follow the **same Markov Chain** from time $t$ to $0$ during denoising due to stitching. We have included this additional explanation in Appendix D.3.2 of the revised manuscript.
>
> The bias-variance formalism we presented is an alternate way of looking at the tilt introduced by P-GRAFT. We briefly describe it below:
>
> **Variance:** We theoretically and empirically show that the expected reward conditional variance is low at small values of time $t$ given partially denoised $X_t$. Therefore, if we can sample a "good" state $X_t$ at time $t$ with the fine-tuned model, and use the pre-trained base model to denoise $X_t$ to $X_0$, we will obtain high rewards with high probability. Therefore, it is sufficient to sample "good" $X_t$ at a small enough $t$ to ensure large rewards for $X_0$.
>
> **Bias:** As noted by the reviewer, our exposition shows that shaping the law of $X_t$ is easier than shaping the law of $X_0$.
>
> **Practical Relevance:** We note that the practical relevance of P-GRAFT is justified by multiple large scale experiments across tasks, where P-GRAFT achieves **the best performance** compared to other alternatives.
>
> Additionally, we have added Appendix D.3.1 in the revised manuscript, which looks at **two edge cases** to provide an **intuition** of how the distribution induced by the stitched model looks like. In summary, if the intermediate latent has no information regarding the final reward, the stitched distribution is identical to the reference distribution - i.e., there is no tilt. On the other hand, if the intermediate latent deterministically determines the final reward, P-GRAFT induces the same distribution as GRAFT/ entropy-regularized MDP.
>
> **References:**
>
> [1] Information Theory: From Coding to Learning, Yihong Wu and Yury Polyanskiy

---

> ### Author Response · Authors · 2025-11-19
> **Regarding Inverse Noise Correction**
>
> > (writing) this section (i.e. Sec. 5) is not clearly written to the point that I could not fully grasp the presented idea. The problem tackled seems not particularly related with the problems treated in the rest of the paper, and is not sufficiently formalized to properly understand the gains of the proposed methodology. I would strongly suggest to also introduce algorithmic aspects with an intuitive presentation of their workings before/after presenting their implementation.
>
> > What is the core algorithmic intuitive idea for the method introduced in Sec. 5?
>
> Section 5 continues the development of ideas developed in Section 4, namely the effectiveness of fine-tuning at the stage of intermediate latents instead of the final denoised vector. In particular, we look at flow models. Since flow models follow an ODE,  the final vector is already determined by the initial noise sample. Therefore, to make changes to the final sample distribution, it is **sufficient to make changes to the initial noise distribution**.
>
> We make use of this observation to propose an algorithm to **correct for the errors in a pre-trained flow model**.  Since an ODE is reversible, if the flow model is learned perfectly, given a sample from the true data distribution, the reverse ODE will sample from the Normal distribution. However, if the learning is imperfect, the reverse ODE samples from a noise distribution that is not Gaussian. We show that it is sufficient to learn this noise distribution to correct for the errors. Due to the theory developed in Sec. 4, learning this distribution is easier than learning the distribution at the fully denoised level.
>
> The idea is summarized in Figure 2. Noise sampled from Normal distribution, following the forward ODE, results in a learned distribution different from the true data distribution due to imperfections. Consequently, the reverse ODE, when applied to samples from the true data distribution, results in a noise distribution different from the Normal distribution. By learning this new “corrected” noise distribution, and then following the forward ODE (the pre-trained model) one can sample from the true data distribution.
>
> We do believe that these ideas are sufficiently formalized in Section 5. In particular, we discuss the connection with Section 4 in Lines 300-316, the errors associated with flow models and the reverse ODE in Lines 317-371 and the intended objective of the proposed algorithm in Lines 372-377. We follow this with details of the algorithm. Could the reviewer clarify which part of the exposition in Section 5 was unclear and difficult to follow?

---

### Author Response · Authors · 2025-12-03
**Final Remarks**

We thank all the reviewers for their thoughtful reviews. We briefly address the main concerns as highlighted by the reviewers:

**Regarding contributions:**

One of the concerns raised by the reviewers was the novelty of GRAFT. We note that our main algorithmic contributions are P-GRAFT and Inverse Noise Correction. GRAFT is a conceptual framework to unify existing rejection sampling methods (such as RAFT), and additionally allow improvements such as de-duplication. The specific utility of GRAFT in the context of diffusion, namely achieving exact marginal KL regularization implicitly is discussed.  Thus, we believe that an extended discussion of GRAFT is necessary to motivate our algorithmic contributions – i.e, P-GRAFT and Inverse Noise Correction.

Additionally, as far as we are aware, we are the first to demonstrate significant performance improvement using rejection sampling methods (GRAFT) over policy gradient methods in the context of large scale diffusion model fine tuning/ alignment.

**P-GRAFT and the Bias-Variance Tradeoff:**

P-GRAFT is a novel algorithm which achieves strong performance boost in diverse settings for diffusion model based generation (including Text-to-Image and Molecule Generation). Our mathematical justification for P-GRAFT, based on the bias-variance tradeoff, provides a conceptual framework to think about fine-tuning diffusion models beyond the trajectory based regularization seen in prior works. As demonstrated empirically, P-GRAFT consistently outperforms GRAFT, practically justifying the presented mathematical analysis.

**Inverse Noise Correction:**

Our mathematical analysis based on the bias-variance tradeoff for P-GRAFT suggests a powerful 2-step algorithm for flow models, since the initial point determines the final output (0 reward variance case as per P-GRAFT formalism). That is, the intermediate state can be chosen as the initial state itself for flow models. We use this to propose  parameter efficient image generation with rectified flows, achieving significant gains by correcting for errors in the initial noise distribution. We have given a detailed mathematical explanation of how this algorithm is derived and its workings in Section 5, with sufficient formalism.

---

### Meta-Review · Area_Chair_JuNn · 2026-01-13

**Summary:**

The manuscript addresses the question of reward-shaping the distributions captured by diffusion models for downstream applications. Noting the intractability of marginal likelihoods required for policy gradient methods such as PPO, authors seek to draw connections between rejection-sampling based finetuning (RaFT etc, which do not require these marginals) and PPO with a reshaped reward. In doing so, they construct a generalised rejection sampling scheme, and a series of subsequent developments on this to better shape distributions for applications.
An analysis of bias-variance tradeoff is provided in support of the method, and experiments conducted to evaluate the contributions.

Reviews appear quite mixed, with the primary questions relating to clarification of the contribution and exposition, with some nuanced issues around comparison to inference-time fine-tuning methods and control-theoretic methods (from reviewer iasd), and the theory-experiment gap of continuous-time SDE vs DDPM/DDIM (from reviewer xSnM).
The rebuttals from the authors provides what looks like a reasonable response in terms of comparisons needing value functions in some way, or that there could be arguments made that potential gaps due to discretisation error are not too relevant.

The overall picture appears to be borderline: after reviewing the paper fully myself, and going through the reviews and rebuttals, I think there is a case to be made for interesting results, but the way the paper is currently structured and written makes this quite hard to really tease out. I don't mean in the sense that the what is being proposed is not clear, but that the context setting and teasing apart some of the nuances in comparison, and framing of experiments given the theory, could and most likely should be done much better to pass the bar for acceptance.

**Reviewer Concerns:**

I'm not entirely sure the primary reviewer concerns noted in the summary above were covered entirely. At least for xSnM, the reviewer response seems to suggest major questions were addressed, but this was also someone who appears to have kept overall score unchanged (only improved soundness).

The issue of there being some reasonable things being proposed and shown in bits and pieces making it difficult to pull everything together still holds to some extent I believe. The authors have provided some additional narrative changes that help with some context, but it feels like more discussion of iasd's questions (and author responses) should really have featured in the manuscript.

**Reviewer Scores:**

iasd: (3,1,2; 2) -> (3,1,3; 3)
sMVE: (2,2,2; 2) -> (3,2,2; 3)
xSnM: (2,3,2; 6) -> (3,3,2; 6)

---

### Decision · Program_Chairs · 2026-01-26

Accept (Poster)